# A dynamic charge-charge interaction modulates PP2A:B56 substrate recruitment

Xinru Wang[1†], Dimitriya H Garvanska[2†], Isha Nasa[3], Yumi Ueki[2], Gang Zhang[2], Arminja N Kettenbach[3,4], Wolfgang Peti[1], Jakob Nilsson[2*], Rebecca Page[1*]

[1]Department of Chemistry and Biochemistry, University of Arizona, Tucson, United States; [2]The Novo Nordisk Foundation Center for Protein Research, Faculty of Health and Medical Sciences, University of Copenhagen, Copenhagen, Denmark; [3]Department of Biochemistry and Cell Biology, Geisel School of Medicine at Dartmouth, Hanover, United States; [4]Norris Cotton Cancer Center, Geisel School of Medicine at Dartmouth, Medical Center Drive, Lebanon, United States

**Abstract** The recruitment of substrates by the ser/thr protein phosphatase 2A (PP2A) is poorly understood, limiting our understanding of PP2A-regulated signaling. Recently, the first PP2A:B56 consensus binding motif, LxxIxE, was identified. However, most validated LxxIxE motifs bind PP2A:B56 with micromolar affinities, suggesting that additional motifs exist to enhance PP2A:B56 binding. Here, we report the requirement of a positively charged motif in a subset of PP2A:B56 interactors, including KIF4A, to facilitate B56 binding via dynamic, electrostatic interactions. Using molecular and cellular experiments, we show that a conserved, negatively charged groove on B56 mediates dynamic binding. We also discovered that this positively charged motif, in addition to facilitating KIF4A dephosphorylation, is essential for condensin I binding, a function distinct and exclusive from PP2A-B56 binding. Together, these results reveal how dynamic, charge-charge interactions fine-tune the interactions mediated by specific motifs, providing a new framework for understanding how PP2A regulation drives cellular signaling.

**\*For correspondence:**
jakob.nilsson@cpr.ku.dk (JN);
rebeccapage@email.arizona.edu
(RP)

[†]These authors contributed equally to this work

**Competing interests:** The authors declare that no competing interests exist.

## Introduction

Protein serine/threonine phosphatase 2A (PP2A) is one of the defining members of the ser/thr phosphoprotein phosphatase (PPP) family that, together with protein phosphatase 1 (PP1), regulates over 90% of all ser/thr dephosphorylation events in eukaryotic cells (*Eichhorn et al., 2009*). PP2A is also recognized as a tumor suppressor because its inactivation by small molecules, viral proteins or endogenous inhibitors leads to tumor formation (*Bialojan and Takai, 1988*; *Pallas et al., 1990*; *Ruvolo, 2016*; *Williams et al., 2019*). Cellular and biochemical studies further confirmed this by demonstrating the role of PP2A in orchestrating mitotic events, cell apoptosis, metabolism and many other fundamental cellular signaling pathways (*Nilsson, 2019*; *Reid et al., 2013*; *Reynhout and Janssens, 2019*; *Wlodarchak and Xing, 2016*). In spite of our advances in understanding PP2A signaling, there is a comparative lack of information about how substrates are specifically recruited to PP2A.

The PP2A holoenzyme is a heterotrimer, composed of a scaffolding subunit A (PPP2R1), a regulatory subunit B (PPP2R2-PPP2R5) and a catalytic subunit C (PPP2C) (*Cho and Xu, 2007*; *Xu et al., 2008*; *Xu et al., 2006*). The A and C subunits form the PP2A core enzyme. Although this core enzyme is relatively invariant, the variable and interchangeable regulatory B subunits result in a diversity of distinct PP2A holoenzymes. There are four known families of B subunits, B55 (B'), B56 (B', PR61), PR72 (B''), and PR93 (B'''), that differ in both their primary sequences and tertiary

structures. Moreover, within each B subunit family, the existence of multiple isoforms and splicing variants further increases the overall number of potential PP2A holoenzymes (*Eichhorn et al., 2009*). Thus, it is the highly variable B subunits that determine PP2A holoenzyme substrate specificity. However, how the B subunits mediate substrate binding at a molecular level is only now beginning to become clear.

Recent structural and biochemical studies showed that PP2A, like other PPPs (i.e., PP1, PP2B/Calcineurin (CN)) bind conserved short linear motifs (SLiMs) found within the intrinsically disordered region (IDR) of its substrates and regulators (*Tompa et al., 2014*; *Van Roey and Davey, 2015*; *Wang et al., 2016*). These SLiMs are disordered in their free form but typically become ordered upon binding structured domains (*Dyson and Wright, 2002*). Crystal structures of ordered SLiMs in complex with PPPs show that these recognition events are, in most cases, driven by the interaction of hydrophobic SLiM residues that bind deep hydrophobic pockets on the PPPs (*Peti et al., 2013*). Single SLiMs typically bind to their cognate PPPs with moderate affinities (*Li et al., 2007*). However, the existence of multiple SLiMs within a single regulator/substrate, coupled with post-translational modifications like phosphorylation, can greatly alter their affinity for their respective PPP (*Bajaj et al., 2018*; *Grigoriu et al., 2013*; *Kumar et al., 2016*; *Nasa et al., 2018*).

More recently, it has been discovered that IDPs can also form high affinity complexes that are dynamic; that is, in which the IDPs simultaneously retain their intrinsic structural disorder (*Borgia et al., 2018*). In these cases, binding is typically driven by electrostatics (*Borgia et al., 2018*; *Hendus-Altenburger et al., 2019*; *Luo et al., 2016*). That is, multiple residues of opposite charge facilitate binding while the lack of a requirement for deep pockets allows the IDPs to retain their inherent dynamics. This emerging paradigm for biomolecular interactions may explain, in part, why hundreds of IDPs have long stretches of positive (lys, arg) and and/or negative (asp, glu) residues (*Borgia et al., 2018*). Namely, that they dynamically contribute to intermolecular interactions.

Recently, the first PP2A:B56 specific SLiM, the LxxIxE motif, was identified (*Hertz et al., 2016*; *Wang et al., 2016*; *Wu et al., 2017*). Proteins containing validated LxxIxE motifs bind B56, an all α-helical heat repeat protein, in a deep, highly conserved hydrophobic pocket. Most LxxIxE motifs bind PP2A:B56 with moderate affinities (low micromolar $K_D$), similar to those observed for other PPP-specific SLiMs. However, for PP1 and CN, enhanced affinities are achieved by exploiting avidity; namely, regulators and substrates contain two or more distinct SLiMs which, together, result in tight affinities for their cognate PPP (*Choy et al., 2014*; *Grigoriu et al., 2013*). Thus far, no such enhancement has been identified for PP2A:B56 as only the LxxIxE motif has been identified. Whether PP2A:B56 uses a similar mechanism to enhance and or modulate the affinities of LxxIxE containing regulators/substrates is thus a major outstanding question.

## Results

### A subset of PP2A:B56 specific substrates depend on a conserved acidic patch in B56 for PP2A:B56 binding

An analysis of the amino acid conservation among 150 distinct B56 sequences shows that the residues that comprise the concave surface of B56 are exceptionally conserved (*Figure 1A*). This conserved region includes the deep, hydrophobic binding pocket that specifically binds the LxxIxE SLiM (*Figure 1A*; *Hertz et al., 2016*; *Wang et al., 2016*). However, the conserved region is much larger, suggesting that regions adjacent to the LxxIxE pocket might also contribute to regulator/substrate binding. An examination of the electrostatic potential of the same region led to the identification of a surface adjacent to the LxxIxE pocket that is not hydrophobic, but instead is highly negatively charged (*Figure 1B*). This surface is defined by multiple acidic residues that are perfectly conserved in B56, both among its various isoforms and throughout evolution (*Figure 1A*, *Figure 1—figure supplement 1*).

To identify if PP2A:B56 substrates/regulators are affected by mutating the conserved B56 acidic patch, we mutated two acidic amino acids in this negatively charged area to arginines (B56α$_{2R}$: E335R/D338R, B56γ$_{2R}$: E310R/D313R; *Figure 1C*) and identified proteins associated with YFP-B56α/γ or YFP-B56α/γ$_{2R}$ from mitotic HeLa cells using quantitative label free mass spectrometry (MS). The MS analysis shows that a subset of the LxxIxE-containing B56 interactors are regulated specifically by the acidic patch for B56 binding in both isoforms. This includes the mitotic regulators KIF4A,

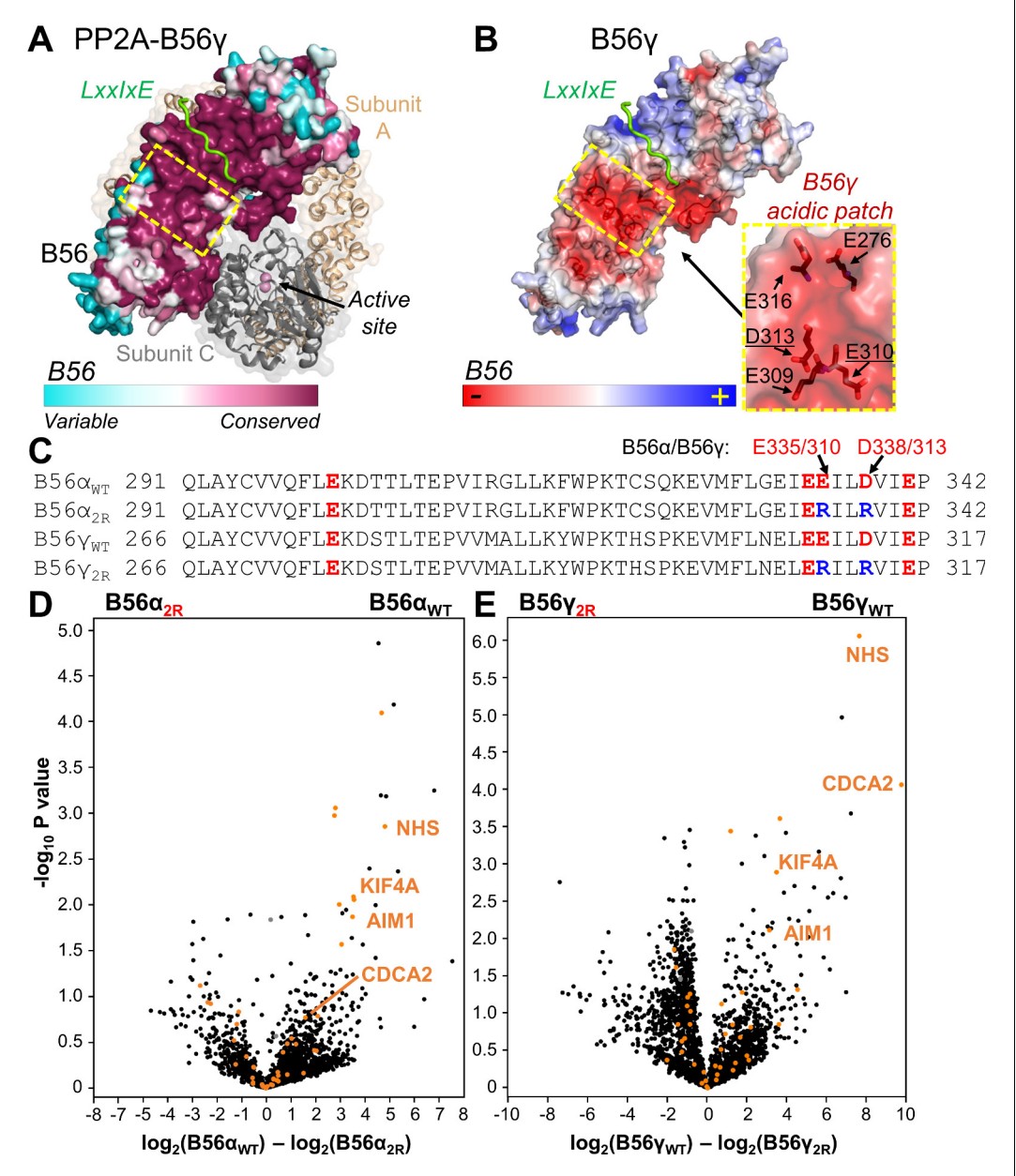

**Figure 1.** The PP2A:B56 holoenzyme uses a conserved acidic patch to bind to B56-specific interactors. (A) PP2A:B56γ holoenzyme (PDBID 2NPP): scaffolding subunit A (beige) and catalytic subunit C (grey; bound metals shown as pink spheres) illustrated as cartoons with transparent surfaces. The regulatory B subunit, B56, is shown as a surface and colored by sequence conservation. An LxxIxE peptide (RepoMan: [588]PLL*p*SP*IPE*LPE[598]; *p* indicates residue is phosphorylated) bound to B56γ is shown in green (PDBIDs 5SW9 and 2NPP superimposed using B56). The location of the conserved acidic patch in B56 (see B) is highlighted with a dashed, yellow square. (B) The B56γ:LxxIxE complex (PDBID 5SW9) colored according to electrostatic potential; LxxIxE peptide is in green. The B56 residues that comprise the conserved acidic patch (yellow dashed square) are shown as sticks and labeled (right; residues mutated in the '2R' mutants underlined). (C) Sequences of B56α and B56γ that comprise the acidic patch, with the acidic residues colored red. The B56 '2R' variants indicate the acidic residues mutated to arginine 'R'. (D) Volcano plot representing the mass spectrometry-identified proteins co-purifying with YFP-B56α versus YFP-B56α$_{2R}$ (E335R/D338R) from mitotic HeLa cells expressing YFP-B56α or YFP-B56α$_{2R}$. PPP2R1A (PP2A regulatory subunit A, α isoform), PPP2CA (PP2A catalytic subunit, α isoform) are labeled in grey. Predicted and confirmed LxxIxE containing proteins (*Hertz et al., 2016*; *Wang et al., 2016*) are highlighted in orange. Four of the six most significantly affected LxxIxE containing B56 interactors selected for further study [NHS, AIM1, CDCA2 (RepoMan) and KIF4A] are labeled. (E) Same as (D) except for YFP-B56γ versus YFP-B56γ$_{2R}$ (E310R/D313R).

The online version of this article includes the following source data and figure supplement(s) for figure 1:

**Source data 1.** List of B56 acidic patch dependent interactors.

*Figure 1 continued on next page*

Figure 1 continued

**Figure supplement 1.** Sequence alignment of the B56 acidic patch.
**Figure supplement 2.** The impact of altering the B56 acidic patch in mitotic progression.

RepoMan (CDCA2), Nance-Horan syndrome protein (NHS) and absent in melanoma 1 protein (AIM1), (ratio WT/2R > 9, p-value<0.05; *Figure 1D,E*, *Figure 1—source data 1*). While the perturbation of these interactions is not sufficient to perturb B56 function in supporting mitotic timing in HeLa cells (*Figure 1—figure supplement 2A,B,C*), it does reveal that the B56 acidic patch is a key binding determinant for a subset of LxxIxE containing PP2A-B56 interactors (*Figure 1D,E*).

To delineate the contribution of the acidic patch, we investigated the molecular site in KIF4A, RepoMan, NHS and AIM1 that is responsible for binding the B56 acidic patch. We reasoned that the B56 acidic patch interacts with a complementary basic patch in B56 interactors. Analysis of the primary sequences of these regulators highlighted the presence of a conserved basic charged rich region within ~15 amino acid N-terminal to an established LxxIxE motif, which we defined as a basic patch (*Figure 2A*, *Figure 2—figure supplement 1*). To measure the contribution of each basic patch to B56 binding, we used isothermal titration calorimetry (ITC). The data showed that mutating the KIF4A basic patch (*bpm*, *b*asic *p*atch *m*utant: $^{1208}KKK^{1210}$ to AAA) reduced the affinity of $KIF4A_{1192-1232}$ for B56γ by ~4 fold (*Figure 2B,C,D*, *Table 1*, *Figure 2—figure supplement 2A,B*). Similarly statistically significant reduced affinities were observed when the basic patch motif of RepoMan, NHS and AIM1 were mutated (CDCA2/RepoMan $^{563}RKKK^{566}$ to AAAA; NHS $^{1618}RCR^{1620}$ to ACA; AIM1

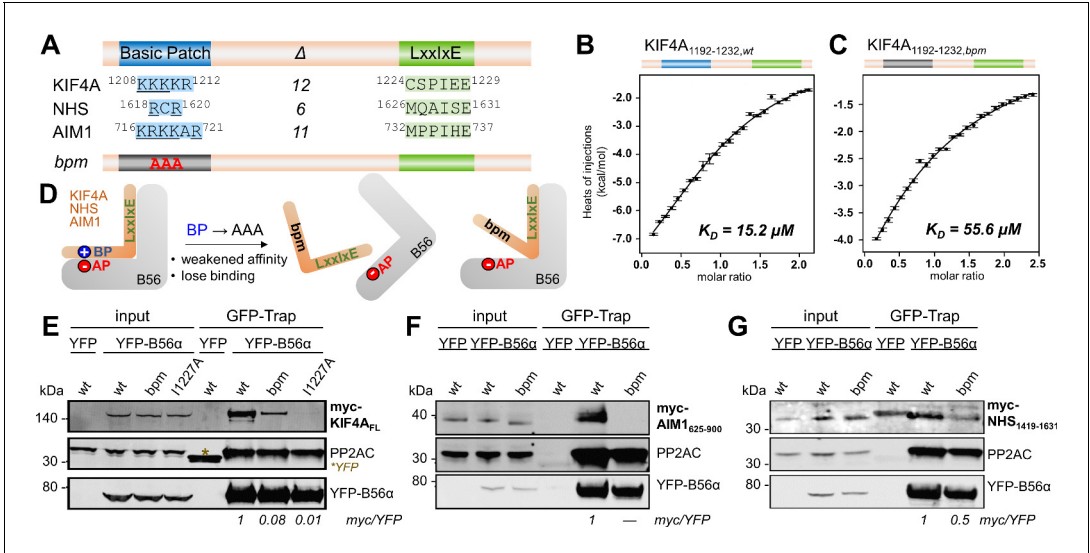

**Figure 2.** KIF4A binds to B56 via a conserved basic patch and an LxxIxE motif. (**A**) B56 interactors with the basic patch (blue) and LxxIxE motif (green) sequences shown; Δ indicates the number of residues between the basic patch and the LxxIxE motif. (**B**) Binding isotherm of WT $KIF4A_{1192-1232}$ with B56γ. (**C**) Binding isotherm of $KIF4A_{1192-1232,bpm}$ ($^{1208}KKK^{1210}$ to AAA) with B56γ. (**D**) Cartoon representation of the effect of mutating the basic patch (BP) of the bp-dependent interactors on their interaction with B56 (AP, acidic patch). (**E**) Immunoprecipitation of YFP-B56α from cells stably expressing YFP-B56α and transfected with the indicated myc-tagged full-length KIF4A variants; asterisk indicates YFP, which was used as a control. The amounts of myc-$KIF4A_{FL}$ co-purified with YFP-B56α were normalized to the band intensity of YFP. The wt is set to 1. (**F**) Immunoprecipitation of YFP-B56α from cells stably expressing YFP-B56α and transfected with the indicated myc-tagged $AIM1_{625-900}$ variants. The amounts of myc- $AIM1_{625-900}$ co-purified with YFP-B56α were normalized to the band intensity of YFP. (**G**) Immunoprecipitation of YFP-B56α from cells stably expressing YFP-B56α and transfected with the indicated myc-tagged $NHS_{1419-1631}$ variants. The amounts of myc-$NHS_{1419-1631}$ co-purified with YFP-B56α were normalized to the band intensity of YFP.

The online version of this article includes the following figure supplement(s) for figure 2:

**Figure supplement 1.** The basic patches of PP2A-B56 basic patch-specific interactors are conserved throughout evolution.
**Figure supplement 2.** ITC thermograms for various PP2A-B56 interactors (WT and bpm) with B56γ (WT and acidic patch mutants).
**Figure supplement 3.** Mutating the basic patch or the LxxIxE motif of KIF4A reduces KIF4A:B56 binding.

**Table 1.** Isothermal titration calorimetry (ITC) measurements between B56γ and KIF4A.

| $B56\gamma_{12\text{-}380}$ | Titrant | $K_D$ (μM)* | ΔH (kcal/mol) | TΔS (kcal/mol) |
|---|---|---|---|---|
| WT | KIF4A[‡] WT | 15.2 ± 0.1 | −11.7 ± 0.7 | −5.1 ± 0.7 |
| WT | $KIF4A_{bpm}$[†] (K1208A/K1209A/K1210A) | 55.6 ± 16.8 | −11.5 ± 2.2 | −5.6 ± 2.4 |
| WT | $KIF4A_{LE}$ (C1224L/S1225E) | 0.32 ± 0.01 | −10.0 ± 0.1 | −2.0 ± 0.1 |
| WT | $KIF4A_{LE,PE}$ (C1224L/S1225E/A1231P/H1232E) | 0.10 ± 0.01 | −13.1 ± 1.1 | −3.6 ± 1.1 |
| WT | $KIF4A_{LE,PE,bpm}$ (K1208A/K1209A/K1210A/C1224L/ S1225E/A1231P/H1232E) | 0.22 ± 0.02 | −10.7 ± 0.1 | −1.6 ± 0.3 |
| E310R D313R | $KIF4A_{LE,PE}$ (C1224L/S1225E/A1231P/H1232E) | 0.19 ± 0.01 | −11.7 ± 0.1 | −2.5 ± 0.1 |
| E276R E310R D313R | $KIF4A_{LE,PE}$ (C1224L/S1225E/A1231P/H1232E) | 0.21 ± 0.01 | −8.2 ± 0.4 | −0.9 ± 0.3 |
| WT | RM[§] WT | 0.13 ± 0.01 | −6.0 ± 0.1 | 3.3 ± 0.1 |
| WT | $RM_{bpm}$ (R563A/K564A/K565A/K566A) | 0.28 ± 0.01 | −7.5 ± 0.1 | 1.4 ± 0.1 |
| WT | NHS[¶]WT | 4.9 ± 0.9 | −17.0 ± 2.2 | −9.8 ± 2.1 |
| WT | $NHS_{bpm}$ (R1618A/R1620A) | 54.5 ± 16.9 | −18.1 ± 2.7 | −12.2 ± 2.5 |
| WT | AIM1[**] WT | 0.80 ± 0.09 | −9.2 ± 0.4 | −0.9 ± 0.5 |
| WT | $AIM1_{bpm}$ (K716A/R717A/K718A/K719A/K721A) | 14.9 ± 1.8 | −8.2 ± 0.2 | −1.6 ± 0.2 |

*All reported measurements are performed with ITC buffer (50 mM sodium phosphate pH 7.5, 150 mM NaCl, 0.5 mM TCEP). Errors are from duplicate or triplicate measurements.

[†] bpm, basic patch mutant.

[‡]KIF4A variants, $KIF4A_{1192\text{-}1232}$.

[§]RepoMan (RM) variants, $RM_{533\text{-}603}$.

[¶]NHS variants, $NHS_{1616\text{-}1635}$.

[**]AIM1 variants, $AIM1_{716\text{-}741}$.

$^{716}$KRKKAR$^{721}$ to AAAAAA; *Table 1*, *Figure 2—figure supplement 2C–H*). Together, these data illustrate that the key role of proximally located basic patches for B56 binding.

To determine if this change in affinity also alters B56 binding in cells, we generated myc-tagged constructs of KIF4A, NHS and AIM1 either as WT or a version where the basic patch was mutated (*bpm*). For KIF4A, we also mutated the LxxIxE motif by mutating the key Ile residue to Ala (I1227A). We then expressed these variants in a cell line stably expressing inducible YFP-B56α and the binding to the myc-tagged variants monitored by affinity purifying YFP-B56α and blotting against the myc-tag. Although all three KIF4A variants (wt, bpm and I1227A) expressed to similar levels, only the WT KIF4A co-purified efficiently with B56α (*Figure 2E*). Similar results were obtained for a myc-tagged basic patch and LxxIxE containing fragment of $KIF4A_{1001\text{-}1232}$ (*Figure 2—figure supplement 3A*), $AIM1_{625\text{-}900}$ (*Figure 2F*) and $NHS_{1419\text{-}1631}$ (*Figure 2G*). Together, these data show that, for a subset of PP2A-B56 interactors, the basic patch motif contributes significantly to B56 binding.

## The binding contribution of the basic patch motif is independent of the strength of the LxxIxE motif

One possible role of the basic patch motif is to selectively enhance B56 affinity for more weakly binding LxxIxE motifs. The LxxIxE motifs have a range of affinities for B56, from stronger (i.e., TLSIKKL(pS)PIIEDDREADH, phosphorylated BUBR1: $K_D$, 0.55 μM) to weaker (i.e., LSTLREQSSQS, Emi2: $K_D$,41 μM) (*Hertz et al., 2016*). The KIF4A LxxIxE motif peptide (CSPIEEAH), like that of Emi2, was previously shown to bind B56 weakly ($K_D$, 32 μM) (*Hertz et al., 2016*). The basic charged motif may not contribute significantly to B56 binding in presence of a tight LxxIxE motif. In order to test this, the KIF4A sequence was mutated to the stronger LxxIxE motif by mutating $^{1224}$CS$^{1225}$ to LE ($KIF4A_{LE}$; the structures of B56γ:LxxIxE complexes show that the 'L, Leu' binds the deep hydrophobic pocket on B56γ while the 'E, Glu' mimics a phosphorylated Ser, which forms multiple salt bridges with B56γ residues H187, R188 [these residues are conserved in all B56 isoforms]). The affinity of

KIF4A$_{1192-1232,LE}$ for B56γ increased 50-fold compared to WT KIF4A (*Table 1*, *Figure 2—figure supplement 2I*; K$_D$ of 0.32 μM). Mutating KIF4A$_{LE}$ residues $^{1231}$AH$^{1232}$ to PE (KIF4A$_{LE,PE}$: the 'P, Pro' positions the 'E, Glu' to form a bidentate salt bridge with B56γ $_{R201}$ *Wang et al., 2016*) further enhanced KIF4A binding (K$_D$ of 0.10 μM; *Table 1*, *Figure 2—figure supplement 2J*). To determine if the basic patch also contributed to B56γ-KIF4A binding in a tight LxxIxE background, we used ITC. Mutating the KIF4A$_{LE,PE}$ basic patch ($^{1208}$KKK$^{1210}$ to AAA) again reduced the binding affinity by ~2 fold (K$_D$ of 0.22 μM; *Table 1*, *Figure 2—figure supplement 2K*), a reduction similar to that observed for WT KIF4A (*Table 1*, *Figure 2B,C*, *Figure 2—figure supplement 2A,B*). This was further confirmed with myc-tagged KIF4A$_{LE}$ and KIF4A$_{LE,bpm}$ variants in cells, showing that while the KIF4A$_{LE}$ variant binds more tightly to B56α compared to WT, mutating the basic patch (KIF4A$_{LE,bpm}$) again reduced binding (*Figure 2—figure supplement 3B*). Together, these data show that the basic patch, together with the LxxIxE motif, are critical for a subset of regulators, including KIF4A, to stably interact with B56, independent of the strength of the LxxIxE motif.

## The basic patch retains its structural disorder when bound to PP2A:B56

To understand how, at a molecular level, the $^{1208}$KKKKR$^{1212}$ basic patch binds B56, we used NMR spectroscopy. An overlay of the 2D [$^1$H,$^{15}$N] HSQC spectra of $^{15}$N-labeled KIF4A$_{1192-1232,LE,PE}$ in the presence and absence of B56γ showed that multiple peaks disappear upon complex formation (*Figure 3A*). Specifically, KIF4A residues 1207 to 1232, which includes the basic patch and the LxxIxE motif, were broadened beyond detection upon binding B56γ. The peaks corresponding to the residues between the two motifs (residues 1213 to 1224) were also broadened beyond detection, indicating this region either is involved in binding or that the conformational freedom of the linker is limited, due to the anchoring of the basic patch and the LxxIxE motifs to B56γ. A crystal structure of the KIF4A$_{1192-1232,LEPE}$:B56γ complex (*Figure 3—figure supplement 1*; *Table 2*) showed that while the LxxIxE motif is well-ordered, electron density corresponding to a single conformation of the basic patch bound to B56 was, as expected, not observed. A crystal structure of the AIM1$_{716-741}$:B56γ complex (*Figure 3—figure supplement 1*; *Table 2*) was similar. Namely, while the AIM1 LxxIxE motif was well-ordered, electron density corresponding to a single conformation of the basic patch bound to B56 was not observed. Together, these data, coupled with the ITC results, show that the KIF4A basic patch interaction with B56 belongs to an emerging class of biomolecular complexes in which one or more partners of the complex retains their structural disorder upon complex binding. That is, the basic charged patch of KIF4A binds B56γ but does so via dynamic, rapidly interchanging conformations even when bound to B56.

The current data suggest that this emerging class of biomolecular interactions is driven almost exclusively by electrostatics (*Borgia et al., 2018*). To confirm that the KIF4A basic patch interacts dynamically with the conserved acidic patch on B56 (*Figure 1B*), we used mutagenesis coupled with NMR spectroscopy. Specifically, the interaction of $^{15}$N-labeled KIF4A$_{1192-1232,LE,PE}$ with four distinct B56γ acidic patch variants was tested: (1) B56$_{2R}$, E310R/D313R, (2) B56$_{2Rb}$, E276R/E316R, (3) B56$_{3R}$, E276R/E310R/E316R and (4) B56$_{4R}$, E276R/D313R/E310R/E316R (*Figure 3B*, *Figure 3—figure supplement 2*). The NMR data showed that the peaks corresponding to the KIF4A basic patch residues are present only with B56 variants with mutated acidic patch residues (*Figure 3B,C*). That is, they no longer interact with these variants of B56 (*Figure 3D*). Consistent with these results, ITC showed that KIF4A binds the B56γ acidic patch variants more weakly (*Table 1*, *Figure 2—figure supplement 2L,M*). Interestingly, mutating only E310R/D313R was sufficient to reduce the binding affinity to the same extent as mutating the basic patch in KIF4A (K$_D$: 0.19 ± 0.01 μM and 0.21 ± 0.01 μM for B56γ$_{2R}$:KIF4A$_{LEPE}$, and B56γ:KIF4A$_{LEPE,bpm}$, respectively, *Table 1*, *Figure 2—figure supplement 2L*); additional mutations (i.e., E276R/E310R/D313R) did not further affect the binding (*Table 1*, *Figure 2—figure supplement 2M*). In agreement with this result, we found that the binding of KIF4A and other bp-containing B56 interactors to B56 in cells was dependent on both an intact acidic patch in B56 (*Figure 3E*, *Figure 3—figure supplement 3*) and an intact basic patch in KIF4A, as even single amino acid substitutions in the KIF4A basic patch lowered binding (*Figure 3F*). This requirement of the acidic patch for binding is consistent with a charge-charge interaction where KIF4A interacts with B56 in a dynamic manner and each amino acid contributes similarly to the overall K$_D$. Together, these data show that the KIF4A basic patch interacts directly with the B56 conserved acidic patch and this interaction is critical for KIF4A binding.

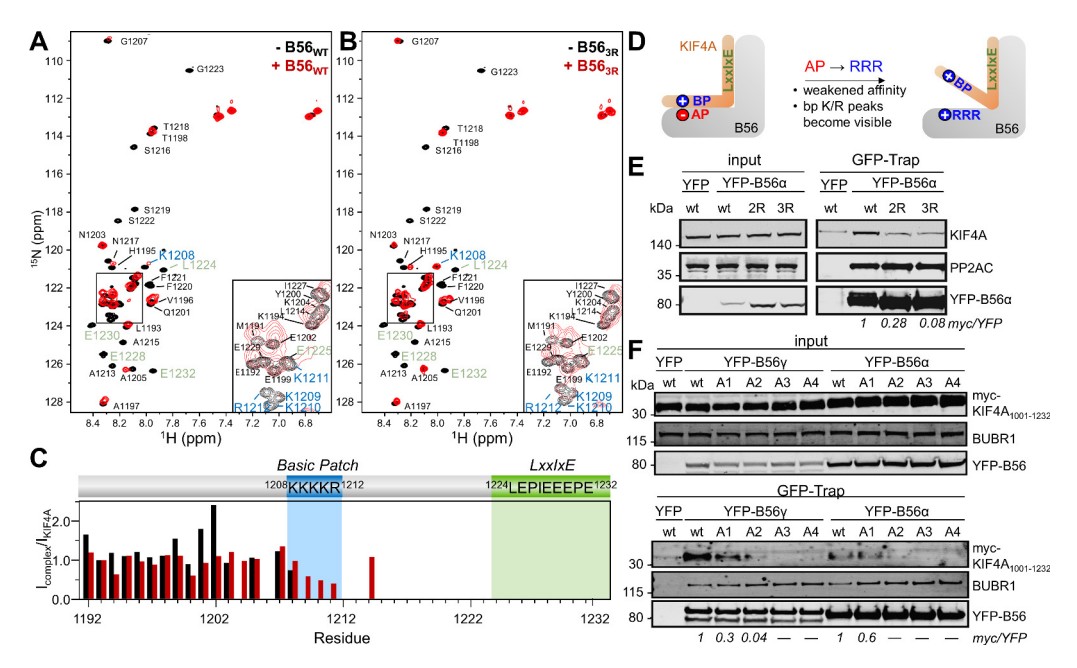

**Figure 3.** The basic patch in B56-specific regulators binds B56 via a dynamic charge-charge interaction. (**A**) Overlay of the 2D [$^1$H,$^{15}$N] HSQC spectra of $^{15}$N-labeled KIF4A$_{1192-1232,LE,PE}$ in the presence (red) and absence (black) of B56γ (1:1 ratio); basic patch and LxxIxE residues labeled blue and green, respectively. (**B**) Overlay of the 2D [$^1$H,$^{15}$N] HSQC spectra of $^{15}$N-labeled KIF4A$_{1192-1232,LE,PE}$ in the presence (red) and absence (black) of B56γ$_{3R}$ E276R/E310R/E316R (1:1 ratio); basic patch and LxxIxE residues highlighted in blue and green, respectively. (**C**) [$^1$H,$^{15}$N] HSQC peak intensity ratios for spectra shown in A, B (black, red, respectively). (**D**) Cartoon representation of the effect of mutating the acidic patch (AP) of B56 on KIF4A:B56 binding (AP: acidic patch, BP: basic patch). (**E**) Immunoprecipitation of stably expressed YFP-B56α variants (wt, B56α$_{2R}$: E335R/D338R, and B56α$_{3R}$ E301R/E335R/D338R and probed for endogenous KIF4A, PP2AC (PP2A catalytic subunit) and GFP (YFP-B56α). (**F**) Immunoprecipitation of transiently transfected myc-tagged KIF4A$_{1001-1232}$ C-terminal variants (A1: K1208A; A2: $^{1208}$KK$^{1209}$ to AA; A3: $^{1208}$KKK$^{1210}$ to AAA; A4: $^{1208}$KKKK$^{1211}$ to AAAA) from cells stably expressing YFP-B56α or YFP-B56γ. The amounts of myc-KIF4A co-purified with YFP-B56 were normalized to the band intensity of YFP.

The online version of this article includes the following figure supplement(s) for figure 3:

**Figure supplement 1.** Crystal structures of KIF4A$_{LE,PE}$:B56γ complex and AIM1:B56γ.

**Figure supplement 2.** Mutating the acidic patch of B56γ reduces the binding of the KIF4A basic patch to B56γ.

**Figure supplement 3.** Mutating the acidic patch of B56 reduces KIF4A:B56 and RepoMan:B56 binding in cells.

## KIF4A dephosphorylation by PP2A:B56 requires the KIF4A basic patch

To determine if the basic charge motif in KIF4A affects dephosphorylation of KIF4A T799, a residue phosphorylated by Aurora B kinase during cytokinesis, we generated a T799 phospho-specific antibody (*Bastos et al., 2014*). Strikingly, the observed phosphorylation level of T799 was inversely correlated with affinities of PP2A-B56 for the different KIF4A variants in mitotic cells (*Figure 4A*). Specifically, mutating either the LxxIxE motif (I1227A) or the basic patch ($^{1209}$KKK$^{1211}$ to AAA) resulted in an increase of phosphorylation of T799, as less PP2A-B56 was recruited to counteract the activity of Aurora B kinase. Further, this phenotype was rescued by enhancing the LxxIxE motif binding affinity; namely, introducing the $^{1224}$CS$^{1225}$ to LE mutation in KIF4A$_{bpm}$ (KIF4A$_{LE,bpm}$) increased the amount of PP2A recruited and, in turn, the amount of KIF4A dephosphorylated. Together, these data show that the dephosphorylation of KIF4A T779 by PP2A-B56 requires the basic patch as the PP2A-B56 dephosphorylation efficacy is directly correlated with PP2A-B56 affinity.

## KIF4A chromosome targeting and PP2A-B56 binding are mutually exclusive as both KIF4A functions strictly require the basic patch

KIF4A is a chromosome-binding kinesin that is important for maintaining normal chromosome architecture during cell division (*Mazumdar et al., 2004*). To determine if the KIF4A basic patch has a mitotic function, we performed RNAi complementation assays in cells where we depleted KIF4A and hKid and then induced the expression of the different YFP-tagged KIF4A variants (*Figure 4B,C,D*;

**Table 2.** Data collection and refinement statistics.

| | B56:KIF4A$_{LE,PE}$[*,†] | B56:AIM1[*,‡] |
|---|---|---|
| PDB | 6OYL | 6VRO |
| Data collection | | |
| Space group | P 2$_1$ 2$_1$ 2$_1$ | I4 |
| Cell dimensions | | |
| a, b, c (Å) | 53.3, 108.0, 117.8 | 111.0, 111.0, 108.9 |
| A, β, γ(°) | 90, 90, 90 | 90, 90, 90 |
| Resolution (Å) | 39.52–3.15 | 39.26–2.45 |
| $R_{merge}$ | 0.100 (1.104) | 0.091 (1.721) |
| Mean I /σI | 11.5 (1.8) | 12.4 (1.2) |
| Completeness (%) | 96.6 (83.1) | 99.8 (99.4) |
| Multiplicity | 8.2 (7.7) | 7.0 (7.0) |
| CC$_{1/2}$ | 0.999 (0.730) | 0.999 (0.673) |
| | | |
| Refinement | | |
| Resolution (Å) | 39.52–3.15 (3.26–3.15) | 38.88–2.45 (2.54–2.45) |
| No. reflections | 11868 | 24208 |
| $R_{work}$/$R_{free}$ | 0.22 (0.36)/0.24 (0.41) | 0.22 (0.33)/0.23 (0.38) |
| No. atoms | | |
| Protein | 2796 | 2777 |
| Water | 7 | 36 |
| B-factors | | |
| Protein | 66.4 | 70.1 |
| Water | 60.4 | 62.1 |
| RMS deviations | | |
| Bond lengths (Å) | 0.002 | 0.002 |
| Bond angles (°) | 0.54 | 0.54 |
| Ramachandran | | |
| Outliers (%) | 0.3 | 0.9 |
| Allowed (%) | 5.8 | 3.4 |
| Favored (%) | 93.9 | 95.7 |
| Clashscore | 4.3 | 2.7 |

[*]Data was collected from a single crystal.

[†]KIF4A$_{LE,PE}$ [1192]ELKHVATEYQENKAPGKKKKRALASNTSFFSGLEPIEEEPE[1232].

[‡]AIM1 [716]KRKKARMPNSPAPHFAMPPIHEDHLE[741].

[*]Values in parentheses are for highest-resolution shell.

KIF4A and hKid are simultaneously depleted because they have nearly fully redundant functions during mitosis *Wandke et al., 2012*). Depleting both KIF4A and hKid resulted in a strong mitotic delay, with multiple unaligned chromosomes. As expected, this phenotype was fully rescued by complementation with WT YFP-KIF4A. However, mutating the basic patch (bpm) resulted in a non-functional KIF4A and this variant failed to localize to mitotic chromosomes (*Figure 4C*, bottom panel). Further analysis revealed that this defect in KIF4A function due to the bpm was not due to a lack of PP2A-B56 binding, as evidenced by the observation that the I1127A variant fully rescued both the mitotic timing and the chromosome alignment phenotypes (*Figure 4C*, middle panel, *Figure 4D*). This demonstrates that the KIF4A basic patch has a function in mitosis, which is distinct from its role in PP2A-B56 binding.

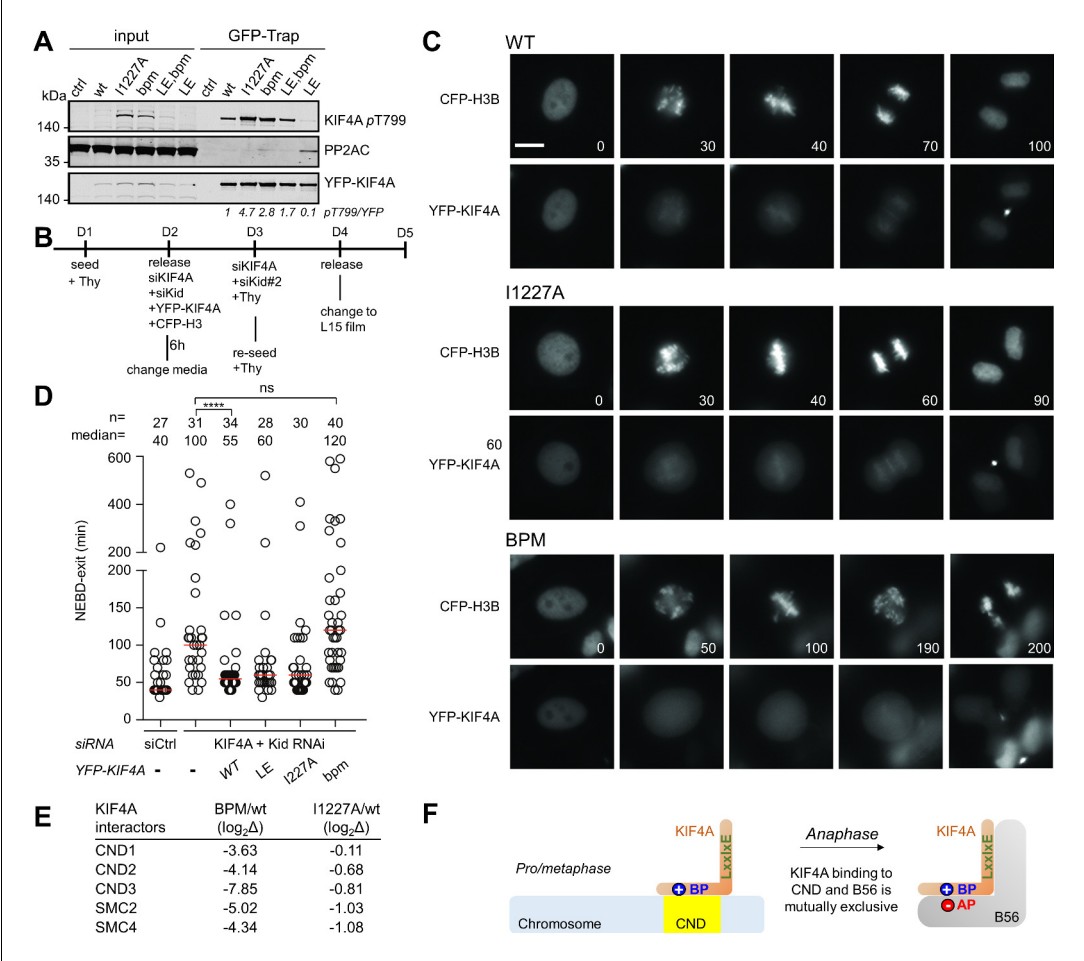

**Figure 4.** The basic patch regulates KIF4A dephosphorylation by PP2A, as well as KIF4A localization in cells. (**A**) The indicated YFP-KIF4A constructs were purified using GFP-Trap and analyzed for phosphorylation by immunoblotting. The T799-phospho signal was normalized to YFP. YFP only was used as a control. (**B**) Endogenous KIF4A was depleted by RNAi and complemented with the indicated YFP-KIF4A variants. (**C**) Live cell imaging of cells expressing YFP-KIF4A variants as they go through mitosis. The beginning of the NEBD was considered as time 0 (min). Bar represents 5 µm. CFP, cyan fluorescent protein. (**D**) Quantification of mitotic duration. Circles represent single cells. The number of cells and median (red line) times are indicated from at least two independent experiments. Mann-Whitney test was used to determine the p-values indicated. **** p<0.0001; * p<0.05; ns, not significant. (**E**) The mass spectrometry-identified condensin complex associated proteins co-purifying with YFP-KIF4Awt versus KIF4A$_{bpm}$ or KIF4A$_{I1227A}$ from mitotic HeLa cells stably expressing YFP-KIF4A variants. (**F**) The binding of chromosome and B56 to KIF4A is mutually exclusive because both binding events strictly require the basic patch.

The online version of this article includes the following source data for figure 4:

**Source data 1.** Separate excel.

To determine if additional proteins bind specifically to the KIF4A basic patch, we purified the different KIF4A mutants from mitotic arrested cells and identified the interacting proteins using MS. These data showed that the abundance of all components of the condensin I complex were strongly reduced in the KIF4A bpm variant (*Figure 4E*, *Figure 4—source data 1*), data that are consistent with the observation that this variant does not localize to chromosomes (*Figure 4C*, bottom panel). In contrast, the abundance of the components of the condensin I complex with the I1227A variant was unaffected, consistent with the ability of this variant to properly target chromosomes during mitosis (*Figure 4C*, middle panel). These data confirm that the basic patch has a second, independent function; namely, it targets KIF4A to chromosomes by binding condensin I. Because we show that both condensin I and PP2A-B56 binding strictly require the basic patch, KIF4A cannot bind both proteins simultaneously, that is, condensin I and PP2A-B56 binding by KIF4A is mutually exclusive

(*Figure 4F*). These data reveal how additional motifs in PP2A-B56 substrates can modulate PP2A-B56 binding to control phospho-dependent signaling in cells.

## Discussion

The traditional view of protein binding is one in which the interacting proteins have well-defined, complementary interfaces (*Lee and Richards, 1971*). However, an emerging mode of binding is that in which one or both proteins exhibit different degrees of disorder in the bound complex (*Berlow et al., 2015*). In particular, the role of highly dynamic, charge-charge interactions that lack well-defined complementary interfaces are becoming increasingly recognized for their central roles in biomolecular interactions and, in turn, a diversity of biological processes like signaling (*Borgia et al., 2018*). The advantage of such a dynamic interaction is that it facilitates fast and responsive regulation. Further, the associated conformational fluctuations also provide ready access for enzymes that mediate post-translational modifications. Because IDPs are not only widely prevalent in eukaryotic genomes (*Brown et al., 2002*), but also unusually enriched in charged amino acids (*Habchi et al., 2014*), the emerging view is that electrostatically-driven dynamic protein:protein interactions are critical for many biological functions.

Here we describe a novel dynamic, charge-charge interaction for a major protein phosphatase, PP2A-B56, which significantly contributes to our understanding of the diversity of mechanisms used by this phosphatase to select its substrates. This novel interaction is mediated by a patch of basic residues that dynamically bind a highly conserved acidic surface present on all B56 isoforms and therefore constitutes a novel pan PP2A-B56 binding motif (*Figure 5*). Charge-charge interactions are generally weak (low µM to mM), yet are becoming increasingly recognized for their importance in increasing the binding affinity of protein:protein interactions, in part by lowering entropy (*Bertran et al., 2019*; *Borgia et al., 2018*; *Luo et al., 2016*; *Sharma et al., 2015*). Further, their

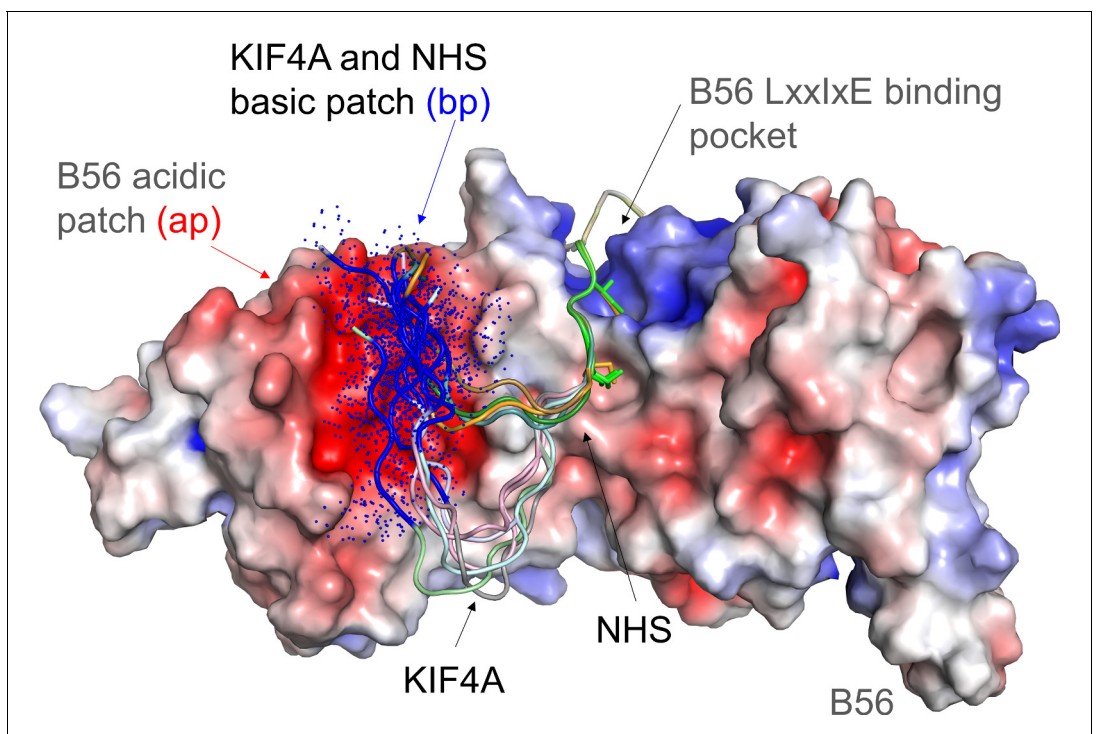

**Figure 5.** Model of the dynamic interaction between the KIF4A basic patch (BP) and the B56 acidic patch. B56 is shown as an electrostatic surface with KIF4A and NHS shown as cartoons. The LxxIxE sequences of KIF4A and NHS bind B56 in a single conformation in the LxxIxE binding pocket (NHS sequence in this pocket modeled using the KIF4A structure, PDBID). As can be seen by these models (generated using COOT and PYMOL), the KIF4A (KKKKR) and NHS (RCR) basic patches (bp, colored dark blue) are optimally positioned to interact dynamically with the B56 acidic patch (ap, red). The dots reflect that these sequences do not adopt a single conformation, but instead retain their intrinsic disorder when bound to the acidic patch.

emerging role in regulating the protein:protein interactions of PPPs with their substrates and regulators is only now beginning to be fully understood. For example, the unrelated B55 subunit also recognizes basic stretches of residues in substrates (*Cundell et al., 2016*) but clearly the substrates for PP2A-B55 and PP2A-B56 are distinct. Thus, additional interactions confer specificity. In the case of PP2A-B56 the binding affinity provided by the basic patch motif is insufficient for B56 binding and requires the presence of an LxxIxE motif, which is specific for B56, in the interactor as well. Thus, the function of the dynamic basic patch is to modulate and enhance the interactions mediated by the LxxIxE motif. It may also facilitate LxxIxE binding by providing an initial docking interaction after which the stronger LxxIxE stabilizes substrate binding; this is consistent with the view that long-range dynamic electrostatic interactions may function as an initial 'tether', after which the specific hydrophobic interactions that, in this case, define the LxxIxE-B56 complex, stabilize binding (*Borgia et al., 2018*). Finally, we show that the presence and contribution of the basic patch is important for a number of interaction partners to bind PP2A-B56 underscoring its relevance and generality.

In the case of KIF4A, we also show that the basic patch motif has a second function. Namely, it is strictly required for KIF4A association with chromatin via condensin I binding; see also *Poser et al. (2019)*. Because both KIF4A functions, binding to PP2A-B56 and condensin, strictly require the basic patch, the binding of both proteins to KIF4A is mutually exclusive. This implies that KIF4A only binds PP2A-B56 upon dissociation from chromosomes, consistent with the reported function of the KIF4A-PP2A-B56 complex in regulating the anaphase central spindle (*Bastos et al., 2014*). Since basic patch motifs resemble nuclear localization motifs (NLSs) recognized by importin α/β, the basic patch motifs would also be inaccessible for PP2A-B56 binding during nuclear transport (*Lange et al., 2007*). Thus, the accessibility of basic patch motifs provide an additional layer of regulation that shapes the PP2A-B56 interaction and dephosphorylation landscape in cells. Together, these discoveries are advancing our understanding of how, at a molecular level, PP2A-B56 engages its substrates and how these interactions are subject to additional regulation via competition for other binding interactions, which has broad implications for understanding cellular signaling.

## Materials and methods

### Sequence alignment

The ConSurf server (using 150 unique B56 sequences with the lowest E values) was used to calculate the conservation scores illustrate in *Figure 1A* (*Ashkenazy et al., 2016*). Clustal Omega (*Madeira et al., 2019*) was used to generate sequence alignments in *Figure 1C*, *Figure 1—figure supplement 1* and *Figure 2—figure supplement 2*. The following species are included in *Figure 1—figure supplement 1*: *Homo sapiens* (human), *Mus musculus* (mouse), *Gallus gallus* (chicken), *Danio rerio* (fish), and *Xenopus laevis* (frog), *Candida albicans* (Candida), *Arabidopsis thaliana* (*A. thaliana*), *Chlamydomonas reinhardtii* (Algae).

### Cloning and expression

Human B56γ1 (B56γ$_{12-380}$ and B56γ$_{31-380}$) was sub-cloned into the pRP1b vector (*Peti and Page, 2007*). B56γ$_{12-380}$ and B56γ$_{31-380}$ were expressed in *E. coli* BL21 (DE3) (Agilent). Cells were grown in Luria Broth in the presence of selective antibiotics at 37°C to an OD$_{600}$ of ~0.8, and expression was induced by the addition of 0.5 mM isopropyl β-D-thiogalactoside (IPTG). Induction proceeded for ~18–20 hr at 18°C prior to harvesting by centrifugation at 6000 *xg*. Cell pellets were stored at −80°C until purification. Human KIF4A (KIF4A$_{1192-1232}$) and RepoMan (RepoMan$_{560-603}$) were sub-cloned into a MBP-fusion vector. Mutants were generated using the QuikChange site-directed mutagenesis kit (Agilent) and sequence confirmed. KIF4A$_{1192-1232}$ and RepoMan$_{560-603}$ variants were expressed in *E. coli* BL21 (DE3-RIL) (Agilent). Cells were grown in Luria Broth in the presence of selective antibiotics at 37°C to an OD$_{600}$ of ~0.6, and expression was induced by the addition of 0.5 mM isopropyl β-D-thiogalactoside (IPTG). Induction proceeded for 5 hr at 37°C prior to harvesting by centrifugation at 6000 *xg*. Cell pellets were stored at −80°C until purification.

Mammalian expression constructs were cloned into pcDNA5/FRT/TO and derivatives of this vector using standard cloning procedures. Point mutations were introduced by whole plasmid PCR

using complementary primers containing the mutations and confirmed by full sequencing of insert. YFP tagged versions of B56α and B56γ were described previously (*Kruse et al., 2013*).

## Generation of stable cell lines

The generation of stable HeLa cell lines expressing constructs under the control of a doxycycline-inducible promoter was carried out as previously described (*Hein and Nilsson, 2014*).

## Cell culture

HeLa-FRT stable cell lines and HeLa cells were passaged in DMEM supplemented with 10% fetal bovine serum (FBS, HyClone) and 1% penicillin-streptomycin (Life Technologies). Protein expression was induced by the addition of doxycycline (Clontech Laboratories) at final concentration of 4 ng/ml.

## Transfection and RNAi

For biochemical experiments cells were transfected with 1.5 µg plasmid and Lipofectamine 2000 (2 µl/ml) for 5 hr, where applicable.

## B56 RNAi rescue

B56 RNAi depletion was done using the following protocol: 250 µl transfection mix with 2 µl siRNA Max (Invitrogen) and 1 µl siRNA oligo (stock concentration 10 µmol) in Optimem (Life Technologies) was added to 750 µl Optimem in 6-well dishes with cells. After 5–6 hr of treatment, FBS was added (10%) until the medium was changed the next day. B56 isoforms were depleted using Dharmacon oligonucleotides against B56α (UGAAUGAACUGGUUGAGUAUU), B56γ (GGAAGAUGAACCAACG UUAUU), B56δ (UGACUGAGCCGGUAAUUGUUU) and B56ε (GCACAGCUGGCAUAUUGUAUU) and used at final concentration of 20 nmol (80 nmol total for all four isoforms). Luciferase (Sigma) was used as control. In live cell experiments, YFP-B56 expressing Hela FRT cell lines were depleted by RNAi 48 hr and 24 hr prior to filming. For KIF4A live cell experiments cells were treated with RNAi 48 hr prior to imaging. YFP-tagged proteins were induced before imaging by the addition of 0.5 ng/ml Doxycycline.

## KIF4A RNAi rescue

HeLa cells were seeded in 6-well plates and synchronized by thymidine the day before transfection. Double RNAi against Kid (CAAGCUCACUCGCCUAUUGUT) and KIF4A (GAAAGATCCTGGC TCAAGA) were performed at 48 and 24 hr before live cell imaging analysis. 800 ng of YFP-tagged wild type KIF4A and mutant plasmids were co-transfected with 30 ng of CFP-Histone3 and RNAi oligos in the first RNAi. Thymidine was added again in the second transfection. After the second RNAi, the cells were re-seeded into 8-well chamber dishes (Ibidi). Cells were released from thymidine in the morning for live cell imaging, which was performed 5 hr later on a DeltaVision Microscope (GE Healthcare).

## Microscopy

Cells were seeded in an 8-well chamber dishes (Ibidi) the day before imaging. After changing the medium to L-15 (Life Technologies) supplemented with 10% FBS and 0.5 ng/ml Doxycycline (where applicable) cells were imaged on a DeltaVision Elite microscope (GE Healthcare) using a 40 × oil immersion objective (1.35 NA, WD 0.10). DIC and YFP channels where imaged with 5 min intervals for 17 hr, taking three z-stacks 5 µm apart. SoftWork software (GE Healthcare) was used for data analysis. Cells expressing within a certain YFP expression window was all analyzed while cells expressing high levels of YFP tagged proteins was excluded from the analysis. ImageJ (NIH) was used to extract still images.

## Immunoprecipitation

Cells were seeded and transfected with myc-tagged constructs (where applicable) on the day after seeding. Following 24 hr thymidine (2.5 mM) block, cells were released into Nocodazole (200 ng/ml) overnight. Inducible cell-lines (YFP-B56, YFP-KIF4A or YFP) were induced 24 hr prior collection with 4 ng/ml Doxycycline. Mitotic cells were collected by shake-off. Cells were lysed in low salt lysis buffer (50 mM Tris pH 7.4, 50 mM NaCl, 1 mM EDTA, 1 mM DTT, 0.1% vol/vol NP40), supplemented with

protease and phosphatase inhibitors (Roche) for 25 min on ice. Lysates were cleared for 15 min at 20,000 xg. Lysates were incubated with 10 µl pre-equilibrated GFP-trap beads (Chromotek) for 1 hr at 4°C and rotation. Beads were washed three times with lysis buffer and eluted in 25 µl 2x LSB (Life Technology) supplemented with 10% β-mercaptoethanol.

## Western blotting

Following SDS–PAGE separation, gels were blotted onto Immobilion FL membrane (Millipore). Membranes were incubated with the indicated primary antibody and subsequently with IRDye 800 or 680 secondary antibodies (Li-Cor). Membranes were scanned using the Odyssey Sa imaging system (Li-Cor) and quantification was carried out using the Odyssey Sa Application software (Li-Cor). Representative images from at least two independent experiments is shown in all figures.

## Antibodies

The following antibodies were used for western-blotting: KIF4A rabbit (Cat# A301-074A; 1:1000, Bethyl laboratories), KIF4A T799 rabbit (raised against peptide CLRRR(pT)FSLT, 1:100, Moravian biotechnology), PP2A-C mouse monoclonal (Cat# 05–421, 1:2,000, Merck), C-myc mouse monoclonal (Cat# SC-40, 1:1000, Santa Cruz), GFP rabbit (raised against full length GFP, 1:10000, Moravian Biotechnology), GFP mouse monoclonal (Cat# 11814460001, 1:1000, Roche), B56α mouse monoclonal (Cat# 610615, 1:1000, BD Transduction Laboratories), BubR1 mouse monoclonal (raised against BubR1 TPR domain, 1:1000, BRIC monoclonal antibody facility), CDC2A rabbit (Cat# HPA030049, 1:1000, Sigma).

## Protein purification

B56γ cell pellets were resuspended in ice-cold lysis buffer (50 mM Tris pH 8.0, 0.5 M NaCl, 5 mM imidazole, 0.1% Triton X-100 containing EDTA-free protease inhibitor tablet [Sigma]), lysed by high-pressure cell homogenization (Avestin C3 Emulsiflex) and centrifuged (35,000 xg, 40 min, 4°C). The supernatant was loaded onto a HisTrap HP column (GE Healthcare) pre-equilibrated with Buffer A (50 mM Tris pH 8.0, 500 mM NaCl and 5 mM imidazole) and was eluted using a linear gradient of Buffer B (50 mM Tris pH 8.0, 500 mM NaCl, 500 mM imidazole). Fractions containing the protein were pooled and dialyzed overnight at 4°C (50 mM Tris pH 8.0, 500 mM NaCl) with TEV protease to cleave the $His_6$-tag. The cleaved protein was incubated with $Ni^{2+}$-NTA beads (GEHealthcare) and the flow-through collected. The protein was concentrated and purified using size exclusion chromatography (SEC; Superdex 75 26/60 [GE Healthcare]) pre-equilibrated in ITC buffer (50 mM sodium phosphate pH 7.5, 150 mM NaCl, 0.5 mM TCEP) or crystallization buffer (20 mM HEPES pH 7.8, 500 mM NaCl, 0.5 mM TCEP). Fractions were pooled, concentrated to designated concentration for experiments or stored at −80 °C. $KIF4A_{1192-1232}$ was purified similarly except that it was heated at 80°C for 10 min and centrifuged (15,000 xg, 10 min, 4°C) prior to SEC purification (SEC buffer: 20 mM HEPES pH 7.8, 500 mM NaCl, 0.5 mM TCEP).

## Crystallization and structure determination

Pooled $B56γ_{31-380}$ (hereafter, B56) in SEC buffer was concentrated and combined with $KIF4A_{1192-1232,LE,PE}$ or AIM1 peptide ($^{716}$KRKKARMPNSPAPHFAMPPIHEDHLE$^{741}$, Bio-Synthesis Inc), in the same buffer at a 1:5 molar ratio to a final concentration of 10 mg/ml. Crystals of the complex were identified in 0.1 M HEPES pH 7.75, 0.8 M LiCl and 8% PEG8K (B56: $KIF4A_{1192-1232,LE,PE}$) or 0.1 M Tris pH 8.0, 0.9 M LiCl and 9% PEG6K (B56: AIM1) using vapor diffusion hanging drops. Crystals were cryo-protected by a 30 s soak in mother liquor with 30% glycerol and immediately flash frozen. Data were collected at SSRL beamline 12.2 at 100 K and a wavelength of 0.98 Å using a Pilatus 6M PAD detector. The data were processed using XDS (*Kabsch, 2010*), Aimless (*Evans and Murshudov, 2013*) and truncate (*French and Wilson, 1978*). The structures of the complexes were solved by molecular replacement using Phaser (*Adams et al., 2010*), using B56 (PDBID 5K6S) as the search model (*Wang et al., 2016*). A solution was obtained in space group $P2_12_12_1$ (B56: $KIF4A_{1192-1232,LE,PE}$) or I4 (B56: AIM1); strong electron density for both peptides was visible in the initial maps. The initial models of the complex were built without the peptide using AutoBuild, followed by iterative rounds of refinement in PHENIX and manual building using Coot (*Emsley and Cowtan, 2004*). The

peptide coordinates were then added followed by iterative rounds of refinement in PHENIX and manual building using Coot. Data collection and refinement details are provided in *Table 2*.

## Isothermal titration calorimetry

SEC was performed to polish B56$\gamma_{12-380}$, RepoMan, KIF4A and exchange into ITC Buffer (50 mM sodium phosphate pH 7.5, 150 mM NaCl, 0.5 mM TCEP). Purified or purchased peptides were titrated into B56$\gamma_{12-380}$ (30 µM) using an Affinity ITC SV microcalorimeter at 25°C (TA Instruments). Data were analyzed using NITPIC, SEDPHAT and GUSSI (*Scheuermann and Brautigam, 2015*; *Zhao et al., 2015*).

## Nuclear magnetic resonance spectroscopy

NMR data were recorded at 283 K using a Bruker Neo 600 MHz ($^1$H Larmor frequency) NMR spectrometer equipped with a HCN TCI active z-gradient cryoprobe. NMR Measurements of KIF4A were recorded using either $^{15}$N- or $^{15}$N,$^{13}$C-labeled protein at a final concentration of 0.1 or 3 mM in NMR buffer (20 mM sodium phosphate pH 6.8, 200 or 50 mM NaCl, 0.5 mM TCEP) and 90% $H_2O$/10% $D_2O$. Unlabeled B56$\gamma_{12-38}$ and $^1$H,$^{15}$N-labeled KIF4A complex was formed via co-SEC (20 mM sodium phosphate pH 6.8, 200 mM NaCl, 0.5 mM TCEP). The sequence-specific backbone assignments of KIF4A and variants were achieved using 3D triple resonance experiments including 2D [$^1$H,$^{15}$N] HSQC, 3D HNCA, 3D HN(CO)CA, 3D HN(CO)CACB and 3D HNCACB. All NMR data were processed using Topspin 4.0.5 and analyzed using Cara. NMR chemical shifts have been deposited in the BioMagResBank (BMRB: 27913).

## Mass spectrometry

Pulldowns were performed in triplicates and analyzed by SDS gel electrophoresis followed by label-free LC-MS/MS on a Q-Exactive Plus quadrupole Orbitrap mass spectrometer (ThermoScientific) equipped with an Easy-nLC 1000 (ThermoScientific) and nanospray source (ThermoScientific) as previously described (*Petrone et al., 2016*). Peptides were resuspended in 5% methanol/1.5% formic acid and loaded on to a trap column (1 cm length, 100 µm inner diameter trap packed with ReproSil $C_{18}$ AQ 5 µm 120 Å pore beads (Dr. Maisch, Ammerbuch, Germany)) vented to waste via a microtee and eluted across a fritless analytical resolving column (35 cm length, 100 µm inner diameter fused silica packed with ReproSil $C_{18}$ AQ 3 µm 120 Å pore beads) pulled in-house (Sutter P-2000, Sutter Instruments, San Francisco, CA) with a 60 min gradient of 5–30% LC-MS buffer B (LC-MS buffer A: 0.0625% formic acid, 3% ACN; LC-MS buffer B: 0.0625% formic acid, 95% ACN). The Q-Exactive Plus was set to perform an Orbitrap MS scan (R = 70K; AGC target = 3e6) from 350 to 1500 Thomson, followed by HCD MS$^2$ spectra on the 10 most abundant precursor ions detected by Orbitrap scanning (R = 17.5K; AGC target = 1e5; max ion time = 75 ms) before repeating the cycle. Precursor ions were isolated for HCD by quadrupole isolation at width = 0.8 Thomson and HCD fragmentation at 26 normalized collision energy (NCE). Charge state 2, 3 and 4 ions were selected for MS$^2$. Precursor ions were added to a dynamic exclusion list +/- 20 ppm for 20 s. Raw data were searched using COMET in high resolution mode (*Eng et al., 2013*) against a target-decoy (reversed) (*Elias and Gygi, 2007*) version of the human (UniProt; downloaded 2/2013, 40482 entries of forward and reverse protein sequences) with a precursor mass tolerance of +/- 1 Da and a fragment ion mass tolerance of 0.02 Da, and requiring fully tryptic peptides (K, R; not preceding P) with up to three mis-cleavages. Static modifications included carbamidomethyl cysteine and variable modifications included: oxidized methionine. Searches were filtered using orthogonal measures including mass measurement accuracy (+/- 3 ppm), Xcorr for charges from +two through +4, and dCn targeting a < 1% FDR at the peptide level. Quantification of LC-MS/MS spectra was performed using MassChroQ (*Valot et al., 2011*) and the iBAQ method (*Schwanhäusser et al., 2011*). Keratin and proteins with a maximum total peptide count of 1 were removed from further analysis. IBAQ quantifications were imported into Perseus (*Tyanova et al., 2016*), and log$_2$ transformed. Missing values were imputed from a normal distribution to enable statistical analysis and visualization by volcano plot. Statistical analysis of protein quantification was carried out in Perseus by two-tailed Student's t-test.

## Data and software availability

All NMR chemical shifts have been deposited in the BioMagResBank (BMRB 27913). Atomic coordinates and structure factors have been deposited in the Protein Data Bank (6OYL, 6VRO). The mass spectrometry proteomics data have been deposited to the ProteomeXchange Consortium (*Vizcaíno et al., 2014*) through the PRIDE partner repository (PXD013886).

## Acknowledgements

This work is supported by grants R35GM119455 and P20GM113132 from the National Institute of General Medical Sciences to ANK, grant R01NS091336 from the National Institute of Neurological Disorders and Stroke and grant R01GM134683 from the National Institute of General Medical Sciences to WP and grant R01GM098482 from the National Institute of General Medical Sciences to RP. Work at the Novo Nordisk Foundation Center for Protein Research is supported by grant NNF14CC0001 and JNI is supported by grant from the Independent Research Fund Denmark (DFF-7016–00086). This research used beamline 12.2 at the Stanford Synchrotron Radiation Lightsource. Use of the Stanford Synchrotron Radiation Lightsource, SLAC National Accelerator Laboratory is supported by the US Department of Energy, Office of Science and Office of Basic Energy Sciences under Contract No. DE-AC02-76SF00515. The SSRL Structural Molecular Biology Program is supported by the DOE Office of Biological and Environmental Research and by the National Institutes of Health, National Institute of General Medical Sciences (including P41GM103393).

## Additional information

### Funding

| Funder | Grant reference number | Author |
| --- | --- | --- |
| National Institute of General Medical Sciences | R35GM119455 | Arminja N Kettenbach |
| National Institute of General Medical Sciences | P20GM113132 | Arminja N Kettenbach |
| National Institute of General Medical Sciences | R01GM098482 | Rebecca Page |
| National Institute of Neurological Disorders and Stroke | R01NS091336 | Wolfgang Peti |
| National Institute of General Medical Sciences | R01GM134683 | Wolfgang Peti |
| Novo Nordisk | NNF14CC0001 | Jakob Nilsson |
| Independent Research Fund Denmark | DFF-7016-00086 | Jakob Nilsson |

The funders had no role in study design, data collection and interpretation, or the decision to submit the work for publication.

### Author contributions

Xinru Wang, Formal analysis, Investigation, Methodology, Writing - original draft, Writing - review and editing; Dimitriya H Garvanska, Formal analysis, Investigation, Methodology, Writing - review and editing; Isha Nasa, Formal analysis, Investigation, Writing - review and editing; Yumi Ueki, Gang Zhang, Investigation; Arminja N Kettenbach, Data curation, Formal analysis, Funding acquisition, Investigation, Writing - review and editing; Wolfgang Peti, Rebecca Page, Conceptualization, Resources, Data curation, Formal analysis, Supervision, Funding acquisition, Validation, Investigation, Writing - original draft, Project administration, Writing - review and editing; Jakob Nilsson, Resources, Data curation, Formal analysis, Supervision, Funding acquisition, Investigation, Writing - original draft, Project administration, Writing - review and editing

## Author ORCIDs
Xinru Wang (iD) https://orcid.org/0000-0001-5994-707X
Isha Nasa (iD) http://orcid.org/0000-0001-7699-795X
Gang Zhang (iD) http://orcid.org/0000-0001-7697-7203
Arminja N Kettenbach (iD) https://orcid.org/0000-0003-3979-4576
Jakob Nilsson (iD) https://orcid.org/0000-0003-4100-1125
Rebecca Page (iD) https://orcid.org/0000-0002-4645-1232

## Decision letter and Author response
Decision letter https://doi.org/10.7554/eLife.55966.sa1
Author response https://doi.org/10.7554/eLife.55966.sa2

## Additional files

### Supplementary files
• Transparent reporting form

### Data availability
All NMR chemical shifts have been deposited in the BioMagResBank (BMRB 27913). Atomic coordinates and structure factors have been deposited in the Protein Data Bank (6OYL, 6VRO). The mass spectrometry proteomics data have been deposited to the ProteomeXchange Consortium (Vizcaíno et al., 2014) through the PRIDE partner repository (PXD013886).

The following datasets were generated:

| Author(s) | Year | Dataset title | Dataset URL | Database and Identifier |
|---|---|---|---|---|
| Page R, Peti W, Wang X | 2020 | The Structure of the PP2A B56 subunit KIF4A complex | http://www.rcsb.org/structure/6OYL | RCSB Protein Data Bank, 6OYL |
| Page R, Peti W, Wang X | 2020 | The structure of the PP2A B56 subunit AIM1 complex | http://www.rcsb.org/structure/6VRO | RCSB Protein Data Bank, 6VRO |
| Kettenback AN | 2020 | Mass spectrometry proteomics data | http://proteomecentral.proteomexchange.org/cgi/GetDataset?ID=PXD013886 | ProteomeXchange, PXD013886 |
| Wang X, Garvanska D, Kettenbach A, Peti W, Nilsson J, Page R | 2020 | Backbone 1H, 13C, and 15N Chemical Shift Assignments for the C-terminal Fragment of a Kinesin KIF4A Variant | http://bmrb.wisc.edu/data_library/summary/index.php?bmrbId=27913 | Biological Magnetic Resonance Data Bank, 27913 |

The following previously published dataset was used:

| Author(s) | Year | Dataset title | Dataset URL | Database and Identifier |
|---|---|---|---|---|
| Kettenbach A | 2019 | A dynamic charge:charge interaction modulates PP2A:B56 interactions | https://massive.ucsd.edu/ProteoSAFe/dataset.jsp?task=9a6d3fdf30db4f6db9243-d3d3cc07058 | MassIVE, MSV0000 83785 |

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
