## [Decision Letter]

**Acceptance summary:**

The concept that multisite interactions are the important for the recognition of substrates by protein kinases and phosphatases is gaining traction. It was already known that the B56 regulatory subunit of the PP2A phosphatase ternary complex can recognize substrates for dephosphorylation through binding to an LXXIXE short motif in the substrate. Here the authors have defined a second substrate recognition mechanism acting in concert with the LXXIXE motif, which involves interaction between a negatively charged groove on the surface of B56 and a basic patch on the substrate protein. They demonstrated that a bidentate substrate interaction involving both the LXXIXE motif and the basic patch is important for PP2A-mediated dephosphorylation of the KIF4A mitotic kinesin and is functionally relevant in mitosis. These findings represent an important advance in our understanding of how PP2A can recognize its phosphoprotein substrates with high affinity and specificity.

**Decision letter after peer review:**

[Editors’ note: the authors submitted for reconsideration following the decision after peer review. What follows is the decision letter after the first round of review.]

Thank you for submitting your work entitled "A dynamic charge:charge interaction modulates PP2A:B56 interactions" for consideration by *eLife*. Your article has been reviewed by three peer reviewers, including Tony Hunter as the Reviewing Editor and Reviewer #1, and the evaluation has been overseen by a Senior Editor.

Our decision has been reached after consultation between the three reviewers. Based on these discussions and the individual reviews below, we regret to inform you that your work will not be considered further for publication in *eLife*.

The reviewers expressed interest in the possibility that the acidic surface patch adjacent to the LpSPIxE SLiM-binding groove in the B56 PP2A holoenzyme regulatory subunit might serve as an additional substrate recognition motif for PP2A/B56 protein phosphatase substrates possessing a basic patch adjacent to the SLiM. The structural and biochemical data you present are largely consistent with the idea that the B56 acidic patch promotes its interaction with KIF4A via its basic patch both in vitro and in vivo and thereby facilitates KIF4A dephosphorylation, although the proposed charge-charge interaction was not evident in the KIF4A/B56 structure you present. The main issue is that you provide no convincing evidence that the B56 acidic patch/KIF4A basic patch interaction is essential for PP2A/B56's or KIF4A biological function in vivo, and this type of evidence would need to be presented for at least one PP2A/B56 substrate of this type. Thus, while the conserved nature of the B56 acidic patch suggests it could be important in PP2A/B56 substrate selection, a stronger example where this is functionally important needs to be established. Whilst the reviewers recognise the potential importance of your findings they concur that as it presently stands your paper does not provide enough new biological insight into PP2A substrate selection to be considered further by *eLife*.

Reviewer #1:

The authors have previously defined LpSPIxE as short linear interacting motif (SLiM) through which the PP2A/B56 protein phosphatase complex targets a subset of its phosphoprotein substrates, such as RepoMan and BubR1. Here, they have extended their investigation of how PP2A-B56 recognizes its substrates. In their original report, three structures of B56 associated with SLiM phosphopeptides from RepoMan and BubR1 were solved to define the mode of LpSPIxE binding to B56. A study of these PP2A/B56/LpSPIxE structures combined with evolutionary analysis of B56 sequences revealed an acidic surface patch adjacent to the hydrophobic pocket that binds the LpSPIxE motif. To investigate a possible functional role of this acidic patch in PP2A/B56 substrate selection, they stably expressed WT and E335R/D338R acidic patch mutant B56α-YFP in HeLa cells, and used MS analysis to compare the repertoire of proteins pulled down with the WT and mutant B56α proteins from nocodazole-arrested mitotic cells. They found several proteins, including RepoMan and KIF4A, that were more weakly associated with the acidic patch B56α mutant. Both B56α and B56γ isoform acidic patch mutants, however, were able to support mitotic progression in HeLa cells depleted of B56α or B56γ, indicating that the mitotic function of PP2A/B56 was not severely perturbed. They then focused on the KIF4A mitotic kinesin as a PP2A/B56 substrate, and investigated the importance of the acidic patch in interaction and dephosphorylation of KIF4A. When KKK in the basic patch N-terminal to the LXXIXE motif was mutated to AAA, they found this reduced the affinity of recombinant KIF4A for recombinant B56 by ~4 fold in vitro. A parallel analysis carried out by mutating the basic patch upstream of the LXXIXE in RepoMan yielded a similar decline in binding to B56. By coexpression in HeLa cells, they observed that unlike basic patch or LXXIXE KIF4A mutants only WT KIF4A was efficiently coprecipitated with B56. By making mutations in the KIF4A LXXIXE motif, they showed that the affinity of the LXXIXE motif for B56 did not affect the contribution of the basic patch to the interaction of the substrate protein with B56 in vitro. Next, they investigated the consequences of mutating the KIF4A basic patch on its T799 phosphorylation status in cells, finding that mutation of either the LXXIXE motif or the basic patch increased pT799 levels, but did not affect the time taken for cells to move from NEBD to anaphase, consistent with the fact that KIF4A depletion also did not affect mitotic timing. However, they did note that KIF4A basic patch mutant failed to associate with mitotic chromosomes, while retaining the ability to localize to the spindle midzone. On this basis, they argue that the KIF4A basic patch will not be accessible to PP2A/B56 when it is associated with chromosomes. Finally, they investigated the dynamics of B56α-KIF4A interaction using NMR perturbation analysis, and found that the basic patch in KIF4A binds the acidic patch of B56 in a dynamic fashion via a charge-charge interaction between the acidic patch n B56 and the basic patch in KIF4A.

The evidence that the acidic patch on the surface of the PP2A B56 regulatory subunit is important for recognition of a subset of PP2A/B56 substrates is reasonably strong, and supported by the structural, biophysical and in vivo data. The disappointing aspect of these studies is that the activity of the single PP2A/B56 substrate they analyzed in depth, i.e. the KIF4A kinesin, that potentially requires the acidic patch for PP2A/B56 recognition, does not exhibit an obvious in vivo phenotype when its basic patch is mutated.

1) Figure 1A, B: The authors need to define the sequence(s) of the LXXIXE peptide(s) displayed in the structure in the figure legend (was this a pSer.Pro-containing peptide).

2) Figure 2C: The binding studies were done with bacterially-expressed MBP-KIF4A_1192-1232_. Presumably this protein was not phosphorylated at S1225 – what difference does the phosphate at S1225 make to binding affinity, and how does the affinity of a pSer-containing motif compare to the affinity when the Ser is changed to the phosphomimic Glu (this is relevant to the B56/KIF4A structure shown in Figure 4—figure supplement 1, where the LE,PE high affinity LXXIXE mutant version of KIF4A was used)?

3) Figure 3B: The myc-KIF4A pT799 bands need quantifying to demonstrate the magnitude of the effects of mutating either the basic patch or the LXXIXE motif on KIF4A dephosphorylation.

4) Figure 4—figure supplement 1: The authors show the crystal structure of the KIF4A basic patch-LEPIEEEPEE motif peptide bound to the B56 HEAT repeat region, where FSGLEPIEEEEPE residues are observed binding between heat repeats 3 and 4, but the atomic level interactions are not shown, and panel with these should be included. A surface representation of the complex like that in Figure 1A, B would also be helpful. Apparently, the KIF4A KKKKR basic patch did not make a stable enough interaction with the B56 acidic patch to be detected in the crystal structure. The NMR perturbation data in Figure 4—figure supplement 1 indicate that K1208 makes the strongest interaction, consistent with the effect of K2208A mutation on binding affinity, while K1209, K1210, K1211 and R1212 interactions are less prominent. Conversely, one would like to know which of the acidic patch residues in B56 are most important. For instance, what is the role of the E335 and D338 B56 residues that were mutated in interacting with the basic patch residues, and which of the five basic residues interact with the four acidic residues (four residues in the B56 acidic patch were mutated to Arg in Figure 4—figure supplement 1)? Can the authors model the basic patch residues into their structure, and, if so, what sort of turn would the intervening sequence between the LXXIXE motif and the basic motif have to make for both motifs to be bound simultaneously (also see point 8).

5) It is not always clear which isoform of B56 was used in different experiments, and this should be indicated e.g. B56γ was used for the crystal studies shown in Figure 4—figure supplement 1.

6) Did the authors try a charge reversal experiment in which they mutated the basic patch in KIF4A to an acidic patch based on their structural information, and test whether this restores KIF4A dephosphorylation mediated by the E335R/D338R acidic patch mutant B56α-YFP in HeLa cells.

7) Could the dynamic basic patch-acidic patch interaction be used as an initial docking interaction between a in a candidate substrate protein that collides with PP2A/B56 and PP2A/B56, which could then allow stabilization of substrate binding by the LXXIXE motif, if one is present adjacent to the basic patch?

8) The number of residues between the LXXIXE motif and the basic motif differs significantly between RepoMan and KIF4A, and this should be discussed. Based on analysis of other PP2A/B56 substrates that might use an analogous basic motif, can the authors deduce rules for how close and how far away the two motifs can be?

9) What does the basic patch of KIF4A interact with on chromosomes – is it an acidic patch on another chromosomal protein or perhaps DNA itself?

Reviewer #2:

The manuscript from groups doing leading work in understanding how serine-threonine phosphatases interact with their substrates. A high affinity interaction site on the 'B56' family of PP2A subunits has previously been identified. Here, they demonstrate a highly conserved acidic patch on the surface of B56 subunits, and mutation of this patch affects the affinity of a few selected substrates of B56 containing PP2A holoenzymes. In the case of the most markedly modified interaction, with KIF4A, the effect of the mutation on affinity is 4-fold, with changes documented with several diverse techniques. The limitation of the impact of this result is that the mutation has biochemical consequences but no detectable effect of the mutants on mitotic timing. The binding affinity effect size is smaller with a second substrate, RepoMan, at only two-fold. A crystal structure of the B56 and KIF4A peptide apparently fails to identify the interaction, which they suggest might be due to the dynamic nature of the interaction. So in the end, there is a small quantitative effect of this acidic patch on the one best substrate, and no biological consequence detected. The findings will be of interest to PP2A aficionados but do not rise to the level of broad biological significance.

Figure 1: There is no table detailing the other proteins that were differentially affected in the pulldown in Figure 1C, D. Shouldn't this data be presented in the appendix? How many of these have SLiM domains and basic patches?

Also, how were the P values calculated and what method was used to correct for multiple comparisons?

There is no alignment of RepoMan identifying its basic patch. Did other interactions from the experiments in Figure 1C, D have basic patches?

The legend for Figure 1C states the 2R mutant B56α is E335R/D338R. Figure 1B shows neither of these as being labelled in the acidic patch. I found this confusing. Is this just a nomenclature/numbering issue? 2NPP and 5SW9 structures both are with B56γ – it would be helpful to label it as such in Figure 1A, and identify the specific amino acids mutated in Figure 1C, D instead of the non-obvious 2R.

The authors use an unconventional naming criteria for PP2A subunits, adding to the nomenclature confusion in the field. The Aa subunit is PPP2R1A, not 2AAA. 2AAA is meaningless. If they need shorthand, why not just use 2R1A? Similarly, PP2AA should be PPP2CA, and if they need 4 letter shorthand, why not use P2CA?

There is a 4-fold effect of mutating the KIF4A patch; there is a twofold effect of mutating the RepoMan patch. This twofold effect is not claimed by the authors to be significant, and I see no statistical test. Please explain.

Figure 3F: No indication of significance is given on the figure, just noted in legend. Please clarify. The text indicates there is no significant effect of the mutants on mitotic timing. Also, the labelling of the figure is out of register.

Figure 3E: The statement that mutating the basic patch on KIF4A abolishes chromosome association goes beyond the data presented here. Additional assays would be needed to show this was not just a problem with the assay or selection of specific images. And that it was due to the binding to B56 rather than to another chromatin binding partner. This section should temper its conclusions or provide additional data.

Figure 4: the text suggest the NMR data is with full length KIF4A. "^15^N-labeled KIF4A in the presence and absence of B56". However, this may be misleading, as the figure legend and figure suggest a different experiment, a small fragment of KIF4A that is mutated to bind B56 with high affinity. Please be careful in the text to describe this accurately. Please explain in the text why the mutant KIF4A fragment was used, if this is indeed the case. Please help me understand why the results with mutant KIF4A should be applicable to non-mutant KIF4A? Does the dynamic interaction with the basic patch require the high-affinity mutation of KIF4A?

If I understand correctly, the crystal of the peptide of KIF4A with B56 did not resolve the interaction of the basic patch with the conserved acidic groove, thus not providing support for the model.

Results paragraph two: “These mutants were able to support normal mitotic timing in B56 RNAi…” So how biologically important can this patch be? It's confusing to me that they find the mutation alters binding of two mitotic regulators yet there is normal timing?

Reviewer #3:

The manuscript by Wang et al., "A dynamic charge:charge interaction modulates PP2A:B56 interactions", presents some novel observations, about substrate binding to PP2A:B56, with most of the experiments focusing on a single protein, KIF4A. Authors identify a conserved acidic groove on B56 (PPP2R5A-E) and show that it binds a basic patch in KIF4A through interaction studies with wild type proteins and mutants in both the B56 acidic groove and the KIF4A basic patch. NMR analyses further demonstrate the presence of this interaction. The experimental results presented are generally of high quality, and this paper will be of interest to anyone studying PP2A. Main weaknesses are the lack of biological impact of the B56 mutants with disruptions in the acidic groove, lack of in vitro dephosphorylation kinetics, and narrow focus on the KIF4A protein, which limits the authors' ability to make broad conclusions about the significance of this interaction. Also, much of the data presented in the manuscript is in the form of immunoblots which should be quantified, and presented with statistical analysis of the results.

Specific comments:

1) Figure 1: Authors provide very limited analysis/discussion of the interaction studies carried out with B56α and B56γ (WT vs. 2R mutant), mentioning only KIF4A and RepoMan. However, there are many orange dots in Figure 1C, D (proteins containing LxxIxE motifs), some of which also show significantly altered interactions. The story would be significantly strengthened by making this story more general: Are there basic patches present in LxxIxE containing proteins that are disrupted by the 2R mutations as compared to proteins that are not disrupted? For KIF4A the basic patch is 8 amino acids N terminal to LxxIxE. Is this spacing similar in RepoMan? If so, how many other candidates similarly display the presence of a basic patch close to LxxIxE (again vs. proteins whose interaction was not disrupted). Does addition of a strongly basic peptide selectively elute these interactors from B56?

– Furthermore, many proteins that do show significant changes with WT vs. 2R mutants do not have predicted LxxIxE motifs. Authors should comment on this. Are they indirect interactors? Do they have LxxIxE motifs not previously identified? Or another B56 binding motif?

– Authors state that the acidic patch in B56 is conserved, but Figure 1—figure supplement 1 shows an alignment of human sequences only. This should be modified to include a broader evolutionary distribution-is this acid groove conserved in lower eukaryotes? Plants?

2) In Figure 2E, F, Myc KIF4A FL containing the basic patch mutations (bpm) shows reduced interaction with B56. However, the KIF4A fragment (1001-1232) is expressed at much higher levels than full length in vivo, contains both basic patch and LxxIxE motifs, and shows complete loss of binding when the basic patch is mutated. Does this indicate the presence of additional interactions between KIF4A and B56 beyond the basic patch and LxxIxE motif? Authors should comment.

3) In Figure 3B a phospho-specific antibody is used to look at steady state phosphorylation levels in vivo, which could be due to altered phosphorylation and/or dephosphorylation. In vitro dephosphorylation assays should be included to determine if the basic patch influences dephosphorylation kinetics.

4) Figure 3E: Authors state that mutating the basic patch in KIF4A abolished chrosmosome association but not localization to the midzone, however the images shown are difficult to interpret as there is no co-staining with chromosomal or kinetochore proteins. There is clearly a bright blob that is present in all images except those for the KIF4A basic patch mutant. What does this blob represent? How many cells were analyzed?

---

## [Author Response]

[Editors’ note: the authors resubmitted a revised version of the paper for consideration. What follows is the authors’ response to the first round of review.]

The reviewers expressed interest in the possibility that the acidic surface patch adjacent to the LpSPIxE SLiM-binding groove in the B56 PP2A holoenzyme regulatory subunit might serve as an additional substrate recognition motif for PP2A/B56 protein phosphatase substrates possessing a basic patch adjacent to the SLiM. The structural and biochemical data you present are largely consistent with the idea that the B56 acidic patch promotes its interaction with KIF4A via its basic patch both in vitro and in vivo and thereby facilitates KIF4A dephosphorylation, although the proposed charge-charge interaction was not evident in the KIF4A/B56 structure you present. The main issue is that you provide no convincing evidence that the B56 acidic patch/KIF4A basic patch interaction is essential for PP2A/B56's or KIF4A biological function in vivo, and this type of evidence would need to be presented for at least one PP2A/B56 substrate of this type. Thus, while the conserved nature of the B56 acidic patch suggests it could be important in PP2A/B56 substrate selection, a stronger example where this is functionally important needs to be established. Whilst the reviewers recognise the potential importance of your findings they concur that as it presently stands your paper does not provide enough new biological insight into PP2A substrate selection to be considered further by eLife.

We appreciate the concerns of the reviewers. As described above, we have performed scores of additional experiments and also made key changes to the manuscript figures and text that reveal the broad relevance of this discovery not only for those interested in PP2A or the family of ser/thr phosphoprotein phosphatases specifically, but also phosphorylation signaling and IDP interactions, generally. Our data now show that the basic patch is critical for binding for multiple interactors, revealing the broad relevance of these interactions. Further, our molecular data (ITC, NMR and the crystal structure) demonstrate conclusively that this interaction between KIF4A and B56 is *dynamic*, in which the basic patch retains its structural disorder upon complex formation (this is further supported by a new crystal structure of the AIM1:B56γ complex, which, again, demonstrates that the basic patch retains its intrinsic disorder upon complex formation). Thus, this study of the molecular basis of the KIF4A-B56 interactions is one of the most thoroughly characterized for this novel paradigm of dynamic charge-charge based biomolecular binding. Finally, we also discovered that the KIF4A basic patch is strictly required for condensin binding. Because of this, the binding of KIF4A to condensin and B56 is mutually exclusive, and thus the requirement of this basic patch provides a novel mechanism for controlling KIF4A-B56 binding in both space and time. These results will be of interest for a broad scientific audience.

Reviewer #1:[…]The evidence that the acidic patch on the surface of the PP2A B56 regulatory subunit is important for recognition of a subset of PP2A/B56 substrates is reasonably strong, and supported by the structural, biophysical and in vivo data.

We thank the reviewer for the recognition of the strength of the structural, biophysical and in vivo data of the manuscript.

The disappointing aspect of these studies is that the activity of the single PP2A/B56 substrate they analyzed in depth, i.e. the KIF4A kinesin, that potentially requires the acidic patch for PP2A/B56 recognition, does not exhibit an obvious in vivo phenotype when its basic patch is mutated.

In the first submission, we had analyzed the function of Kif4A in RNAi rescue experiments and, as the reviewer points out, the removal of Kif4A has limited effect on mitotic progression in HeLa cells, consistent with the literature. We have now repeated these experiments and co-depleted Kif4A and hKid that perform partly redundant functions. This co-depletion results in a strong mitotic phenotype with unaligned chromosomes that is fully rescued by Kif4A WT but not Kif4A BPM. We further analyze this function and show that the basic patch in Kif4A plays a dual function in that it is strictly required for binding both to condesin I and PP2A-B56 which is consistent with the defective localization to chromosomes of Kif4A BPM.

1) Figure 1A, B: The authors need to define the sequence(s) of the LXXIXE peptide(s) displayed in the structure in the figure legend (was this a pSer.Pro-containing peptide).

The sequence of the LxxIxE peptide shown in Figure 1A, B has now been added to the Figure 1A legend.

“An LxxIxE peptide (RepoMan: 590PL*L-pS-PIPE*LPE595; *p* indicates residue is phosphorylated) bound to B56γ is shown in green (PDBIDs 5SW9 and 2NPP superimposed using B56).”

2) Figure 2C: The binding studies were done with bacterially-expressed MBP-KIF4A_1192-1232_. Presumably this protein was not phosphorylated at S1225 – what difference does the phosphate at S1225 make to binding affinity, and how does the affinity of a pSer-containing motif compare to the affinity when the Ser is changed to the phosphomimic Glu (this is relevant to the B56/KIF4A structure shown in Figure 4—figure supplement 1, where the LE,PE high affinity LXXIXE mutant version of KIF4A was used)?

This is correct; S1225 is not phosphorylated in the bacterially expressed and purified KIF4A 1192-1232 construct. As stated in the manuscript, the LE,PE high affinity variant was used to ensure the KIF4A affinity for B56 was in a regime suitable for quantitative measurements of affinity when the basic patch is mutated (ITC, NMR). Multiple studies, including those described in this manuscript, have investigated how changes in the LxxIxE sequence alter B56 affinity. First, as we described previously (1), the phosphorylation of S1225 increases the KIF4A affinity for B56 3-fold while changing the first position of the LxxIxE sequence from the native ‘C’ to ‘L’ increases the binding affinity 23-fold. Second, we show here that introducing the phosphomimetic Glu into the more optimal KIF4A sequence (CSPIEE to LEPIEE) results in an increase in B56 affinity to 50-fold over WT. This latter result is described in the manuscript text as follows.

“In order to test this, the KIF4A sequence was mutated to the stronger LxxIxE motif by mutating 1224CS1225 to LE (KIF4ALE; the structures of B56γ:LxxIxE complexes show that the ‘L, Leu’ binds the deep hydrophobic pocket on B56γ while the ‘E, Glu’ mimics a phosphorylated Ser, which forms multiple salt bridges with B56γ residues H187, R188 [these residues are conserved in all B56 isoforms]). The affinity of KIF4A_1192-1232_,LE for B56 increased 50-fold compared to WT KIF4A (Table1, Figure 2—figure supplement 2I; KD of 0.32 μM).”

3) Figure 3B: The myc-KIF4A pT799 bands need quantifying to demonstrate the magnitude of the effects of mutating either the basic patch or the LXXIXE motif on KIF4A dephosphorylation.

This is now reported below the blot. We have added quantifications to all Western blots.

4) Figure 4—figure supplement 1: The authors show the crystal structure of the KIF4A basic patch-LEPIEEEPEE motif peptide bound to the B56 HEAT repeat region, where FSGLEPIEEEEPE residues are observed binding between heat repeats 3 and 4, but the atomic level interactions are not shown, and panel with these should be included. A surface representation of the complex like that in Figure 1A, B would also be helpful.

As requested, we now include a panel illustrating the atomic level interactions of KIF4A with B56. We also include a surface representation of B56:KIF4A interaction to demonstrate the molecular details of this interaction. We have now also determined the structure of the AIM1:B56γ complex (AIM1 exhibits the greatest loss in affinity upon mutation of the basic patch). As with KIF4A, while the density for the AIM1 LxxIxE motif is well ordered, the AIM1 basic patch also retains its intrinsic disorder upon B56γ binding. We have included atomic level interactions of this new complex as well. These figures are included in the supplemental data of the current manuscript (Figure 3—figure supplement 1C, D).

Apparently, the KIF4A KKKKR basic patch did not make a stable enough interaction with the B56 acidic patch to be detected in the crystal structure. The NMR perturbation data in Figure 4—figure supplement 1 indicate that K1208 makes the strongest interaction, consistent with the effect of K2208A mutation on binding affinity, while K1209, K1210, K1211 and R1212 interactions are less prominent. Conversely, one would like to know which of the acidic patch residues in B56 are most important. For instance, what is the role of the E335 and D338 B56 residues that were mutated in interacting with the basic patch residues, and which of the five basic residues interact with the four acidic residues (four residues in the B56 acidic patch were mutated to Arg in Figure 4—figure supplement 1)? Can the authors model the basic patch residues into their structure, and, if so, what sort of turn would the intervening sequence between the LXXIXE motif and the basic motif have to make for both motifs to be bound simultaneously (also see point 8).

The reviewer is correct that the basic patch does not interact with B56 in a manner that results in a *single conformation* that can be readily identified by X-ray crystallography. Rather, our data collectively show that this interaction is *dynamic,* a hallmark of a new paradigm of biomolecular interactions defined by IDP-based charge-charge electrostatic interactions in which the IDPs retain their intrinsic disorder. Namely, the ITC data show that both the B56 acidic and KIF4A basic patch contribute to binding affinity while the NMR spectroscopy data show that the basic residues engage directly with B56 acidic patch. However, the crystal structures show that this interaction is *not* achieved by forming a single stable structure (i.e., with a single KIF4A basic residue interacting exclusively with a single acidic residue on B56). Instead, our data describe a model in which the KIF4A basic residues interact *dynamically* with the B56 acidic residues. To further demonstrate this, we now include additional data in which increasing numbers of the basic residues in the KIF4A basic patch were mutated and the mutated KIF4A variants pulled down using a GFP-trap (Figure 3F). These new data show that even the loss of even a single lysine is sufficient to disrupt B56 binding. Finally, we also determined a second crystal structure: the AIM1:B56γ complex (mutation of the AIM1 basic patch results in an 18-fold reduction in affinity of AIM1 for B56). As observed for the KIF4A:B56γ complex, while the electron density of the AIM1 LxxIxE motif is well ordered, the AIM1 basic patch retains its structural disorder upon B56 binding.

As highlighted in the Discussion, while dynamic charge-charge interactions are generally moderate (low μM to mM), they are becoming increasingly recognized for their importance in increasing the binding affinity of protein:protein interactions, in part, by lowering entropy (2–4). One of the most extreme cases was recently reported in Nature (5), in which two IDPs (histone H1 and its nuclear chaperon prothymosin-α) bind with picomolar affinity while fully retaining their structural disorder; the exceptional binding affinity is due to the large opposite net charge of the two proteins. We also recently discovered that a dynamic charge-charge interaction between NHE1 and a second PPP, calcineurin (CN), determines dephosphorylation specificity for a specific NHE1 phosphosite (6). What we have discovered here is that a subset of B56 substrates also uses dynamic charge-charge interactions to facilitate B56 binding. Further, we show that the modest changes in affinity due the presence of loss of this electrostatic interaction has profound impacts on B56 binding in vivo. Why? Because of the hundreds of *additional* B56 interactors that contain LxxIxE sequences that are competing for the same binding site on B56. That is, in the absence of the electrostatic interaction, KIF4A is displaced from B56 by other LxxIxE-containing interactors and thus, KIF4A is no longer a PP2A-B56 substrate. This result in profound and significant for the PPP field specifically and the signaling field generally. Namely, that a subset of substrates exploit a dynamic charge-charge interaction to enhance PP2A-B56 binding.

We now include a model that illustrates how the dynamic basic patch of these regulators binds the B56 acidic patch (in which the bp retains its intrinsic disorder) and the LxIxxE sequences binds the B56 LxIxxI binding pocket. This is now included in Figure 5.

5) It is not always clear which isoform of B56 was used in different experiments, and this should be indicated e.g. B56γ was used for the crystal studies shown in Figure 4—figure supplement 1.

This has been addressed by adding the isoform tested throughout the figures and text.

6) Did the authors try a charge reversal experiment in which they mutated the basic patch in KIF4A to an acidic patch based on their structural information, and test whether this restores KIF4A dephosphorylation mediated by the E335R/D338R acidic patch mutant B56α-YFP in HeLa cells.

Although an interesting suggestion, we have not tried this.

7) Could the dynamic basic patch-acidic patch interaction be used as an initial docking interaction between a in a candidate substrate protein that collides with PP2A/B56 and PP2A/B56, which could then allow stabilization of substrate binding by the LXXIXE motif, if one is present adjacent to the basic patch?

Yes, this is most certainly a possibility, which is consistent with the view that long-range dynamic electrostatic interactions may function as the initial ‘tether’, after which the specific hydrophobic interactions that define the LxxIxE-B56 complex stabilize binding. The potential role(s) of this newly discovered dynamic electrostatic interaction is now addressed in the Discussion.

“It may also facilitate LxxIxE binding by providing an initial docking interaction after which the stronger LxxIxE stabilizes substrate binding; this is consistent with the view that long-range dynamic electrostatic interactions may function as an initial ‘tether’, after which the specific hydrophobic interactions that, in this case, define the LxxIxE-B56 complex, stabilize binding (5).”

8) The number of residues between the LXXIXE motif and the basic motif differs significantly between RepoMan and KIF4A, and this should be discussed. Based on analysis of other PP2A/B56 substrates that might use an analogous basic motif, can the authors deduce rules for how close and how far away the two motifs can be?

The number of residues between the LxxIxE and basic patches varies depending on substrate. For example, in the substrates analyzed here, the number of residues between the motifs varies between 6 and 12 amino acids. Further, the change in binding affinity when the basic patch is mutated is not linearly correlated with the number of intervening residues Thus, while not exhaustive, our data suggests that the number of intervening residues is variable.

9) What does the basic patch of KIF4A interact with on chromosomes – is it an acidic patch on another chromosomal protein or perhaps DNA itself?

We have performed a series of MS experiments to answer this question. The MS data now shows that KIF4A interacts directly with the chromosomally associated condensin complex and, further, that this interaction strictly requires the KIF4A basic patch. This MS data is now included as Figure 4—source data 2 and also included in Figure 4E. Our new results are consistent with and, more importantly, extend a recent publication from the Barr laboratory (7) that we cite (see also response to reviewer 2’s point 7).

Reviewer #2:The manuscript from groups doing leading work in understanding how serine-threonine phosphatases interact with their substrates. A high affinity interaction site on the 'B56' family of PP2A subunits has previously been identified. Here, they demonstrate a highly conserved acidic patch on the surface of B56 subunits, and mutation of this patch affects the affinity of a few selected substrates of B56 containing PP2A holoenzymes. In the case of the most markedly modified interaction, with KIF4A, the effect of the mutation on affinity is 4-fold, with changes documented with several diverse techniques. The limitation of the impact of this result is that the mutation has biochemical consequences but no detectable effect of the mutants on mitotic timing. The binding affinity effect size is smaller with a second substrate, RepoMan, at only two-fold. A crystal structure of the B56 and KIF4A peptide apparently fails to identify the interaction, which they suggest might be due to the dynamic nature of the interaction. So in the end, there is a small quantitative effect of this acidic patch on the one best substrate, and no biological consequence detected. The findings will be of interest to PP2A aficionados but do not rise to the level of broad biological significance.

We thank the reviewer for their positive comments on our research programs. As described in the Introduction, our previous and new experimental data show unequivocally that this basic patch is present in a subset of key PP2A-B56 substrates. Further, we show that these basic patches are critical for B56 binding both in vivo and in vitro for *multiple* substrates, including KIF4A, RepoMan, AIM1 and NHS. This demonstrates this is a general mechanism for modulating B56 substrate binding.

As we show, even though the in vitro affinity differences of the substrates for B56 with and without the basic patch are small, these differences have profound impacts on PP2A biology and PP2A-B56 recruitment, as evidenced by the loss of PP2A-B56 binding upon mutating the basic patch residues to alanines (see current Figure 3F). The observation that small affinity differences in vitro have significant effects in vivo is an emerging paradigm for all PPP interactions and thus these results are broadly relevant for multiple biological systems. Specifically, we reported one of the first examples in our *eLife* manuscript describing the interaction/regulation of KI-67 and RepoMan with PP1 (8). Specifically, we showed that the affinity of KI-67 for PP1α versus PP1γ differs only 5-fold in vitro. However, in vivo, this modest change in affinity results in absolutely no binding between PP1α and KI-67 (or RepoMan). Rather, it binds exclusively to PP1γ. Why? Because KI-67 is competing with 100s of other regulators for PP1α and this small difference in affinity in vitro allows all other PP1α-specific regulators to ‘outcompete’ KI-67 for PP1α.

It is also critical to also emphasize how important the PPP family of proteins are for understanding how IDPs engage and direct the activity of key signaling proteins. Our collective work over the last nearly two decades are revealing how these key signaling proteins recognize their regulators and substrates via SLiM interactions, how IDP *dynamics* direct PPP dephosphorylation specificity and, as also described here for the first time for PP2A-B56, the role of *dynamic charge-charge* interactions in modulating substrate binding and substrate function (KIF4A). These discoveries are not only relevant for the PPP family, specifically, but for IDP interactions, generally, which is why we believe that this work rises to the level of broad biological significance. We now also make these points explicitly in the Discussion.

Figure 1: There is no table detailing the other proteins that were differentially affected in the pulldown in Figure 1C, D. Shouldn't this data be presented in the appendix? How many of these have SLiM domains and basic patches?

These data were included in the original submission as an excel table (Figure 1—source data 1). The proteins previously identified as bona fide or predicted LxxIxE interactors are indicated in the last two columns of this file (i.e., those identified by Hertz, et al., 2016 (1) and those identified by Wang, et al., 2016 (9)).

Also, how were the P values calculated and what method was used to correct for multiple comparisons?There is no alignment of RepoMan identifying its basic patch. Did other interactions from the experiments in Figure 1C, D have basic patches?

P-values were calculated in Perseus (10) using a two-tailed Student’s T-test. In the first submission, we did not include a correction for multiple comparisons. This is because we are focused on understanding the binding behavior of a specific B56 SLiM containing protein to WT and BPM mutant B56. However, in the revised manuscript, we now include an additional tab in the supplemental table (Figure 1—source data 1) where we correct p-values for multiple comparisons using the Benjamini-Hochberg procedure. This was performed in Perseus.

The sequence alignment of the substrates/regulators tested in the manuscript is now included in Figure 2A and Figure 2—figure supplement 1.

The legend for Figure 1C states the 2R mutant B56α is E335R/D338R. Figure 1B shows neither of these as being labelled in the acidic patch. I found this confusing. Is this just a nomenclature/numbering issue? 2NPP and 5SW9 structures both are with B56γ – it would be helpful to label it as such in Figure 1A, and identify the specific amino acids mutated in Figure 1C, D instead of the non-obvious 2R.

We thank the reviewer for pointing this out. We have now included a full sequence alignment of B56α and B56γ, with the key residues highlighted to demonstrate their conservation, and also labeled these amino acids in the figure directly, to make this information clearer. We also include the sequences for the 2R mutants. These sequences can now be found in Figure 1C.

The authors use an unconventional naming criteria for PP2A subunits, adding to the nomenclature confusion in the field. The Aa subunit is PPP2R1A, not 2AAA. 2AAA is meaningless. If they need shorthand, why not just use 2R1A? Similarly, PP2AA should be PPP2CA, and if they need 4 letter shorthand, why not use P2CA?

We appreciate the reviewers comment and, clearly, this was confusing. We have also used this nomenclature when PP2A is introduced (in the Introduction). We have deleted the gene name of each protein for the labels in Figures 1D and 1E to maintain the focus on B56 substrates and regulators.

“Introduction: The PP2A holoenzyme is a heterotrimer, composed of a scaffolding subunit A (PPP2R1), a regulatory subunit B (PPP2R2-PPP2R5) and a catalytic subunit C (PPP2C) (Cho and Xu, 2007; Xu et al., 2008, 2006).”

There is a 4-fold effect of mutating the KIF4A patch; there is a twofold effect of mutating the RepoMan patch. This twofold effect is not claimed by the authors to be significant, and I see no statistical test. Please explain.

This effect is statistically significant (see Table 1). We apologize for not stating this explicitly in the original submission, which clearly led to confusion. Further, we have augmented our in vitro study by also testing the role of the basic patch for two additional regulators identified using our B56-2R mutants coupled with MS: Aim1 and NHS, which also exhibit statistically significant reductions in B56 binding. These data are reported in Table 1, Figure 2, and Figure 2—figure supplement 2.

Figure 3F: No indication of significance is given on the figure, just noted in legend. Please clarify. The text indicates there is no significant effect of the mutants on mitotic timing. Also, the labelling of the figure is out of register.

We have repeated these experiments, but now we deplete *both* KIF4A and hKid; this is because KIF4A and hKid have a redundant functions during mitosis (11). Our data show that co-depletion results in robust and statistically significant mitotic delay, with multiple unaligned chromosomes. Further, both phenotypes were fully rescued by WT KIF4A. In this background, we then expressed the different KIF4A variants. The data revealed a clear function for the basic patch in supporting KIF4A activity. This is an important new result as, together, our data not only reveal the importance of this motif for PP2A-B56 binding and activity (via KIF4A T799 dephosphorylation) but also for chromosome targeting. Further, because both activities strictly required the basic patch, our data provides the molecular explanation for the inability PP2A-B56 to co-localize with KIF4A along chromosomes; i.e., they cannot bind KIF4A simultaneously. These data are now reported in Figure 4C, D. In particular, Figure 4D replaces the original Figure 3F.

We have confirmed that there are no panels in any current figure that are out of register. We apologize for any initial confusion this may have caused.

Figure 3E: The statement that mutating the basic patch on KIF4A abolishes chromosome association goes beyond the data presented here. Additional assays would be needed to show this was not just a problem with the assay or selection of specific images. And that it was due to the binding to B56 rather than to another chromatin binding partner. This section should temper its conclusions or provide additional data.

We have not selected specific images; rather, all our live cell recordings of Kif4A BPMs show that these variants no longer localize to chromosomes in mitosis. We have now included a chromosome marker in or live cell assays to make this clearer. To further support these data, we have now also performed mass spectrometry analysis of the different KIF4A variants. These data also show that the basic patch of KIF4A is strictly required for binding to condensin, which are consistent with results from a recent paper from the Barr laboratory (7). These data provides the molecular mechanism for why KIF4A BPM variant is not localizing to chromosomes. More importantly however, our results now explain why condensin and PP2A-B56 binding is mutually exclusive, an observation that has remained an unexplained conundrum in the field; namely, both strictly require the BPM for binding. Thus, accessibility to the basic patch provide an additional layer of regulation that shapes the PP2A-B56 interaction and dephosphorylation landscape in cells.

Figure 4: the text suggest the NMR data is with full length KIF4A. "^15^N-labeled KIF4A in the presence and absence of B56". However, this may be misleading, as the figure legend and figure suggest a different experiment, a small fragment of KIF4A that is mutated to bind B56 with high affinity. Please be careful in the text to describe this accurately. Please explain in the text why the mutant KIF4A fragment was used, if this is indeed the case. Please help me understand why the results with mutant KIF4A should be applicable to non-mutant KIF4A? Does the dynamic interaction with the basic patch require the high-affinity mutation of KIF4A?

As requested, the figure, legend and text have been edited to make it explicitly clear which construct was used for the experiments. The need for the higher affinity KIF4A variant is a consequence of the binding regime that is accurately monitored using both ITC and NMR. Here, we are comparing affinities with and without the basic patch. Critically, as we show in the manuscript, the affinity of the KIF4A basic patch for the B56 acidic patch is not altered by changing the affinity of the KIF4A SLiM for B56 (Results section: The binding contribution of the basic patch motif is independent of the strength of the LxxIxE motif). This demonstrates conclusively that the dynamic interaction with the basic patch does *not* require the high affinity mutation. However, by using the high affinity variant, the interaction is in a regime suitable for monitoring binding using NMR spectroscopy.

The in vivo analysis of Kif4A confirms that these results are relevant in the context of a non-mutant (WT) context.

If I understand correctly, the crystal of the peptide of KIF4A with B56 did not resolve the interaction of the basic patch with the conserved acidic groove, thus not providing support for the model.

In fact, this result directly supports our model. Namely, that the charge-charge interaction is dynamic and does not adopt a single, stable conformation (i.e., as one observes for interactions involving deep hydrophobic pockets such as that of the LxxIxE motif with B56). Rather, the IDP basic residues retain their dynamics (adopt multiple, interchangeable conformations) when interacting with the B56 acidic patch. The importance of dynamic charge-charge interactions between IDPs and their targets is emerging as a key mechanism by which these IDPs, which represent ~30% of the human proteome, engage and direct the activity of their interacting proteins. What we have discovered is that this is used by a subset of PP2A-B56 substrates and peptides to enhance their ability to bind B56 and, in turn, the biological processes they regulate.

Results paragraph two: “These mutants were able to support normal mitotic timing in B56 RNAi…” So how biologically important can this patch be? It's confusing to me that they find the mutation alters binding of two mitotic regulators yet there is normal timing?

We have not observed an effect on mitotic timing (NEBD-Anaphase) with the B56 mutants despite the fact that the interaction with KIF4A and RepoMan is strongly reduced. However depletion of KIF4A has no major impact on mitotic timing (data in original Figure 3F and consistent with reported literature) and abolishing the interaction of RepoMan and PP2A-B56 by mutating the LxxIxE motif does not affect the kinetics of chromosome targeting at anaphase (12). Given this data, we do not anticipate that weakening the interaction between PP2A-B56 and KIF4A/RepoMan would affect mitotic timing (NEBD-Anaphase) consistent with what we observe. Our data thus show that the acidic patch on B56 subunits is not required for mitotic timing in HeLa cells but this does not allow one to conclude that the acidic patch is not biological important. Given the exceptional conservation and the fact that other non-mitotic regulators (AIM1, NHS) depend on the acidic patch for binding it is likely that in a model organism mutation of the acidic patch would result in a phenotype but this is beyond the current scope of this manuscript.

Reviewer #3:The manuscript by Wang et al., "A dynamic charge:charge interaction modulates PP2A:B56 interactions", presents some novel observations, about substrate binding to PP2A:B56, with most of the experiments focusing on a single protein, KIF4A. Authors identify a conserved acidic groove on B56 (PPP2R5A-E) and show that it binds a basic patch in KIF4A through interaction studies with wild type proteins and mutants in both the B56 acidic groove and the KIF4A basic patch. NMR analyses further demonstrate the presence of this interaction. The experimental results presented are generally of high quality, and this paper will be of interest to anyone studying PP2A. Main weaknesses are the lack of biological impact of the B56 mutants with disruptions in the acidic groove, lack of in vitro dephosphorylation kinetics, and narrow focus on the KIF4A protein, which limits the authors' ability to make broad conclusions about the significance of this interaction.

We appreciate the reviewer’s view that our work is of “high quality” and will be of interest to anyone studying PP2A. As indicated throughout this response, we have greatly expanded that the number of substrates investigated and now provide the data that allow us to make broad conclusions regarding the importance of the substrate basic patches for B56 binding for this set of substrates.

Also, much of the data presented in the manuscript is in the form of immunoblots which should be quantified, and presented with statistical analysis of the results.

For all Western blots, which are done using Licor technology, we now have indicated the numbers below the blots. To our knowledge there is no appropriate statistical test for these types of experiments at these sample sizes. We can inform the reviewer that a student t-test shows statistical significance for all experiments.

Specific comments:1) Figure 1: Authors provide very limited analysis/discussion of the interaction studies carried out with B56α and B56γ (WT vs. 2R mutant), mentioning only KIF4A and RepoMan. However, there are many orange dots in Figure 1C, D (proteins containing LxxIxE motifs), some of which also show significantly altered interactions. The story would be significantly strengthened by making this story more general: Are there basic patches present in LxxIxE containing proteins that are disrupted by the 2R mutations as compared to proteins that are not disrupted? For KIF4A the basic patch is 8 amino acids N terminal to LxxIxE. Is this spacing similar in RepoMan? If so, how many other candidates similarly display the presence of a basic patch close to LxxIxE (again vs. proteins whose interaction was not disrupted).

These are very important points and we now address this directly in the text. First, it is important to note that not all proteins that show a difference interact *directly* with PP2A-B56. The proteins known to interact with PP2A-B56 directly (KIF4A, RepoMan, NHS, AIM1, among others) are typically 100s-1000s of amino acids long. Thus, in cells, these proteins that bind PP2A-B56 directly are also bound to a host of additional proteins. Thus, some of the observed changes are due to this indirect interaction.

However, there are also multiple proteins already demonstrated to bind PP2A-B56 directly via LxxIxE motifs (1, 9). Thus, as suggested by the reviewer, we have now expanded the number of targets investigated both in vitro and in vivo in order to further demonstrate the broad applicability of these results.

The role of the variable spacing amongst the proteins that bind directly to PP2A-B56 via LxxIxE motifs that also have basic patches is addressed in the response to reviewer 1’s eight point.

Does addition of a strongly basic peptide selectively elute these interactors from B56?

The binding of substrates and regulators to B56 via LxxIxE motifs are dominated by the LxxIxE motifs and thus a strongly basic peptide is unlikely to selectively elute these interactors. The key is that *in cells*, where there are many LxxIxE motif containing substrates/interactors competing for the same site, the loss of the basic patch reduces the binding affinity allowing them to be ‘outcompeted’ by other interactors.

– Furthermore, many proteins that do show significant changes with WT vs. 2R mutants do not have predicted LxxIxE motifs. Authors should comment on this. Are they indirect interactors? Do they have LxxIxE motifs not previously identified? Or another B56 binding motif?

As discussed above, many of these are likely *indirect* interactors, as those interactors already demonstrated to bind directly to PP2A-B56 are known to interact with multiple additional proteins (i.e., RepoMan also binds PP1γ, CDK1, among others, which, in turn, bind additional proteins). We now point this out explicitly in the text. The proteins previously identified to have LxxIxE motifs are indicated in Figure 4—source data 1.

– Authors state that the acidic patch in B56 is conserved, but Figure 1—figure supplement 1 shows an alignment of human sequences only. This should be modified to include a broader evolutionary distribution-is this acid groove conserved in lower eukaryotes? Plants?

This has been included as requested and confirms the conservation of the acidic patch throughout evolution (Figure 1—figure supplement 1B).

2) In Figure 2E, F, Myc KIF4A FL containing the basic patch mutations (bpm) shows reduced interaction with B56. However, the KIF4A fragment (1001-1232) is expressed at much higher levels than full length in vivo, contains both basic patch and LxxIxE motifs, and shows complete loss of binding when the basic patch is mutated. Does this indicate the presence of additional interactions between KIF4A and B56 beyond the basic patch and LxxIxE motif? Authors should comment.

Although we cannot exclude additional contacts, we are quite confident that the difference reflects a detection issue, as the signal for FL KIF4A wt is stronger than KIF4A wt 1001-1232 allowing us to more readily detect the residual binding of the FL KIF4A BPM.

3) In Figure 3B a phospho-specific antibody is used to look at steady state phosphorylation levels in vivo, which could be due to altered phosphorylation and/or dephosphorylation. In vitro dephosphorylation assays should be included to determine if the basic patch influences dephosphorylation kinetics.

Given the strong correlation between the strength of the Kif4A-PP2A-B56 interaction and T799 phosphorylation level we find it most likely that this is due to the amount of PP2A-B56 activity on T799 and not a change in kinase activity. The experiments are done in nocodazole arrested cells and therefore Aurora B activity is constant in all samples.

4) Figure 3E: Authors state that mutating the basic patch in KIF4A abolished chrosmosome association but not localization to the midzone, however the images shown are difficult to interpret as there is no co-staining with chromosomal or kinetochore proteins. There is clearly a bright blob that is present in all images except those for the KIF4A basic patch mutant. What does this blob represent? How many cells were analyzed?

In response to this concern, we have greatly improved the quality of these images in revised manuscript. In addition, the co-localization with a chromatin marker is also now included. These data can be found in Figure 4C.

References

1) E. P. T. Hertz, et al., A Conserved Motif Provides Binding Specificity to the PP2A-B56 Phosphatase. *Mol. Cell***63**, 686–695 (2016).

2) M. T. Bertran, et al., ASPP proteins discriminate between PP1 catalytic subunits through their SH3 domain and the PP1 C-tail. *Nat Commun***10**, 771 (2019).

3) L. Luo, et al., The Binding of Syndapin SH3 Domain to Dynamin Proline-rich Domain Involves Short and Long Distance Elements. *J. Biol. Chem*. **291**, 9411–9424 (2016).

4) R. Sharma, Z. Raduly, M. Miskei, M. Fuxreiter, Fuzzy complexes: Specific binding without complete folding. *FEBS Lett*. **589**, 2533–2542 (2015).

5) A. Borgia, et al., Extreme disorder in an ultrahigh-affinity protein complex. Nature 555, 61–66 (2018).

6) R. Hendus-Altenburger, et al., Molecular basis for the binding and selective dephosphorylation of Na+/H+ exchanger 1 by calcineurin. *Nat Commun***10**, 3489 (2019).

7) E. Poser, R. Caous, U. Gruneberg, F. A. Barr, Aurora A promotes chromosome congression by activating the condensin-dependent pool of KIF4A. *J. Cell Biol*. **219** (2019).

8) G. S. Kumar, et al., The Ki-67 and RepoMan mitotic phosphatases assemble via an identical, yet novel mechanism. *eLife***5** (2016).

9) X. Wang, R. Bajaj, M. Bollen, W. Peti, R. Page, Expanding the PP2A Interactome by Defining a B56-Specific SLiM. Structure **24**, 2174–2181 (2016).

10) S. Tyanova, et al., The Perseus computational platform for comprehensive analysis of (prote)omics data. *Nat. Methods*
**13**, 731–740 (2016).

11) C. Wandke, et al., Human chromokinesins promote chromosome congression and spindle microtubule dynamics during mitosis. *J. Cell Biol.***198**, 847–863 (2012).

12) J. Qian, et al., Cdk1 orders mitotic events through coordination of a chromosome-associated phosphatase switch. *Nat Commun***6**, 10215 (2015).